# Global state and potential scope of investments in watershed services for large cities

Chelsie L. Romulo [1,2,3], Stephen Posner[4,5], Stella Cousins[6], Jenn Hoyle Fair[7], Drew E. Bennett[8], Heidi Huber-Stearns[9], Ryan C. Richards[2,3,10] & Robert I. McDonald[11]

Investments in watershed services (IWS) programs, in which downstream water users pay upstream watershed service suppliers for actions that protect drinking water, are increasing in number and scope. IWS programs represent over $170 million of investment in over 4.3 million ha of watersheds, providing water to over 230 million people. It is not yet fully clear what factors contribute to the establishment and sustainability of IWS. We conducted a representative global analysis of 416 of the world's largest cities, including 59 (14%) with IWS programs. Using random forest ensemble learning methods, we evaluated the relative importance of social and ecological factors as predictors of IWS presence. IWS programs are more likely present in source watersheds with more agricultural land and less protected area than otherwise similar watersheds. Our results suggest potential to expand IWS as a strategy for drinking water protection and also contribute to decisions regarding suitable program locations.

[1] University of Northern Colorado, Greeley, CO, USA. [2] Center for Conservation and Sustainability, Smithsonian Conservation Biology Institute, Washington, DC, USA. [3] Department of Environmental Science and Policy, George Mason University, Fairfax, VA, USA. [4] Gund Institute for Environment, University of Vermont, Burlington, VT, USA. [5] COMPASS, Silver Spring, MD, USA. [6] Department of Natural Resources Management and Environmental Science, California Polytechnic State University, San Luis Obispo, CA, USA. [7] Yale School of Forestry and Environmental Studies, New Haven, CONN, USA. [8] MacMillan Program in Private Lands Stewardship, Ruckelshaus Institute, Haub School of Environment and Natural Resources, University of Wyoming, Laramie, WY, USA. [9] Institute for a Sustainable Environment, University of Oregon, Eugene, OR, USA. [10] Center for American Progress, Washington, DC, USA. [11] Global Cities Program, The Nature Conservancy, Arlington, VA, USA. Correspondence and requests for materials should be addressed to C.L.R. (email: chelsie.romulo@unco.edu)

Urbanization forecasts estimate that six of ten people will live in cities by 2030[1], increasing impacts on natural resources and demand for ecosystem services, both within and beyond city borders. Recent studies predict an increase in the number of large cities vulnerable to water stress from 35 to 45% in the next 25 years[2]. Watershed degradation and water treatment costs have increased throughout the 21st century[2], and existing water governance systems and engineering responses are proving inadequate and unfeasible both environmentally and economically. This is particularly acute in large cities, where large populations and population densities concentrate large numbers of people dependent on the same water supplies. The increasing demand on water resources, including exogenous factors such as those related to accelerated climate change, are driving new management strategies to sustain the provision of watershed services[3–6]. As water supply concerns grow, policies and tools must adapt to these new contexts[7–9].

Management innovations have begun offering solutions to problems pertaining to water scarcity, quality, and threats to availability[10]. Policies that involve payments or investments in watershed services (IWS) are incentive-based investment mechanisms designed to address the provision and enhancement of water-related ecosystem services[11,12]. As IWS have increased in scope, scale, and geography, so has research and monitoring of these policies and the underlying resource issues[13]. In 2015, Forest Trends' Ecosystem Marketplace, which tracks longitudinal global data on market-based environmental programs, reported over 400 programs actively investing in watershed services around the world. These programs are associated with cities of all sizes with a variety of management goals, and total over $25 billion transacted among all programs globally, covering a land area larger than the size of India to date[11]. This represents a marked increase in the number, geographic scale, and funding of IWS from the 127 programs reported in 2010[11]. Empirical evidence of the broad impact of payment for ecosystem services studies shows positive, though often small, impacts on environmental and social outcomes[14]. Here we focus on the presence of IWS programs for cities rather than an evaluation of their impacts.

IWS programs routinely cite multiple motivations for program creation, such as problems of water quality, water availability, and other biophysical, economic, social and cultural reasons[15]. A variety of disciplinary perspectives such as those from economics[16–18], political science[6,19,20], and ecology[21] hypothesize specific factors that influence the establishment of ecosystems services strategies such as IWS. We define these enabling conditions as factors that increase the likelihood of a change in governance approach, strategy, or management regime[22]. Though much of the research on enabling conditions for ecosystem services programs has been theoretical or has only evaluated specific cases, a recent synthesis of the literature on payments for ecosystem services programs identified 24 distinct enabling conditions (Fig. 1).

Taken together, the growth of IWS and the limited understanding of the specific enabling conditions point to a need to evaluate the factors associated with IWS programs. While Fig. 1 shows detailed understanding of conditions potentially enabling IWS programs, these conditions have not been analyzed in aggregate or evaluated for relative importance. This includes evaluating which factors may be generalizable across contexts and which may be context specific. It is critical to understand not just which conditions may be important to consider, but whether one condition may be more or less important than another, and in which contexts. Additionally, given the amount of data that could potentially be important, an aggregate assessment allows practitioners to focus attention on those variables which are most

important rather than spending time collecting and assessing all possible data.

To address the gap in our understanding of conditions that enable IWS in large urban areas, and to test whether there are generalizable conditions that enable IWS, we synthesized a global data set of city water supplies, creating the first assessment of urban IWS programs for 416 of the world's largest cities (population > 300,000). Using a random forest machine learning algorithm, we tested 17 data sets representing 15 of the 24 identified enabling conditions (Fig. 1) to see, when considered in aggregate, which variables are most important for predicting the presence of IWS as a strategy for managing urban drinking water supplies (Supplementary Data 1). Random forest models are a type of machine learning algorithm that consist of many individual decision trees constructed iteratively with random subsets of predictor and dependent variables. Each decision tree predicts the presence or absence of an IWS program for a city and the model ranks all variables according to aggregate prediction performance in the forest of individual trees[23]. The model constructed by the random forest classification technique allows us to rank variables in terms of relative importance for predicting the existence of IWS in a given city. We selected this method specifically for high classification accuracy, and the ability to model complex higher-order interactions and non-linear relationships between predictor variables[24], including variables of different data types and the ability to asses different types of data (nominal, categorical, numerical, etc) in aggregate[23]. On the basis of our analyses of enabling conditions, we also demonstrated how to use our results to identify large cities that are suitable candidates for future IWS programs. Our study of global patterns and enabling conditions for IWS provides guidance to policymakers, planners, conservation practitioners, and researchers to develop and evaluate programs where important enabling conditions already exist and foster favorable conditions in areas where key enabling conditions do not yet exist.

## Results

**Global and regional distribution of IWS**. We focused on cities in the City Water Map database (CWM) developed by McDonald et al.[25], which contains the data on large cities and their above and below-ground water sources[2]. By combining a literature review with the data from annual surveys of IWS programs, we identified 59 large cities with IWS (Appendix A); 53 of these were in the CWM. We improved the representativeness of our city sampling by first using a sample of cities from the CWM database that was stratified according to city size and UNPD geographical region, and then combining this with the data from a comprehensive survey of IWS programs and a review of published literature on IWS programs. Program information was derived from Forest Trends census surveys[15]. As opposed to sampling, Forest Trends aims to conduct a census (a survey of all identified programs), and uses the data from other sources (articles, websites, reports) when an interviewee is not available to complete the survey for the program. Overall, IWS programs represent over $170 million of investment in over 4.3 million ha of watersheds, providing water to more than 230 million people. Analysis included cities in the Americas, Europe, China and SE Asia, Indonesia, and SE Africa (Fig. 2). Seventeen of the 114 countries represented in the CWM had at least one identified IWS program, but the fraction of cities with and without IWS varied by region. Regions with the highest numbers of large CWM cities did not necessarily have the largest proportions of IWS programs (for example, Southern Asia had 86 cities in the CWM, including 73 cities in India, but our research identified only one IWS program that met our criteria, in New Delhi). Some

| Biophysical conditions | Economic conditions | Governance conditions | Sociocultural conditions |
|---|---|---|---|
| • Small resource area | • Significant value of ES | • Presence/absence of intermediaries | • Trust & transparency among actors |
| • Resource location & arrangement | • Low opportunity costs | • Strong capacity among actors | • Stakeholder communication & engagement |
| • Well-defined boundaries of PES system | • Manageable transaction costs | • Influential champion | • Pre-existing market based culture |
| • Existing fundamental ecosystem science and baseline data | • Defining ES as an economic good or service | • Strong existing institutions | • Participant willingness |
| • Linkages between ES provision and management practices | • Economic growth | • Secure land tenure & property type | • Proximity of actors to each other |
| • Clear threat or risk to ES | | • Fit of governance structure with scale of PES | • Large/small number of actors |
| | | • Multiple/single PES objectives | |

**Fig. 1** Framework for enabling conditions for payments for ecosystem services (PES) programs identified from the literature (Huber-Stearns et al., Fig. 1)

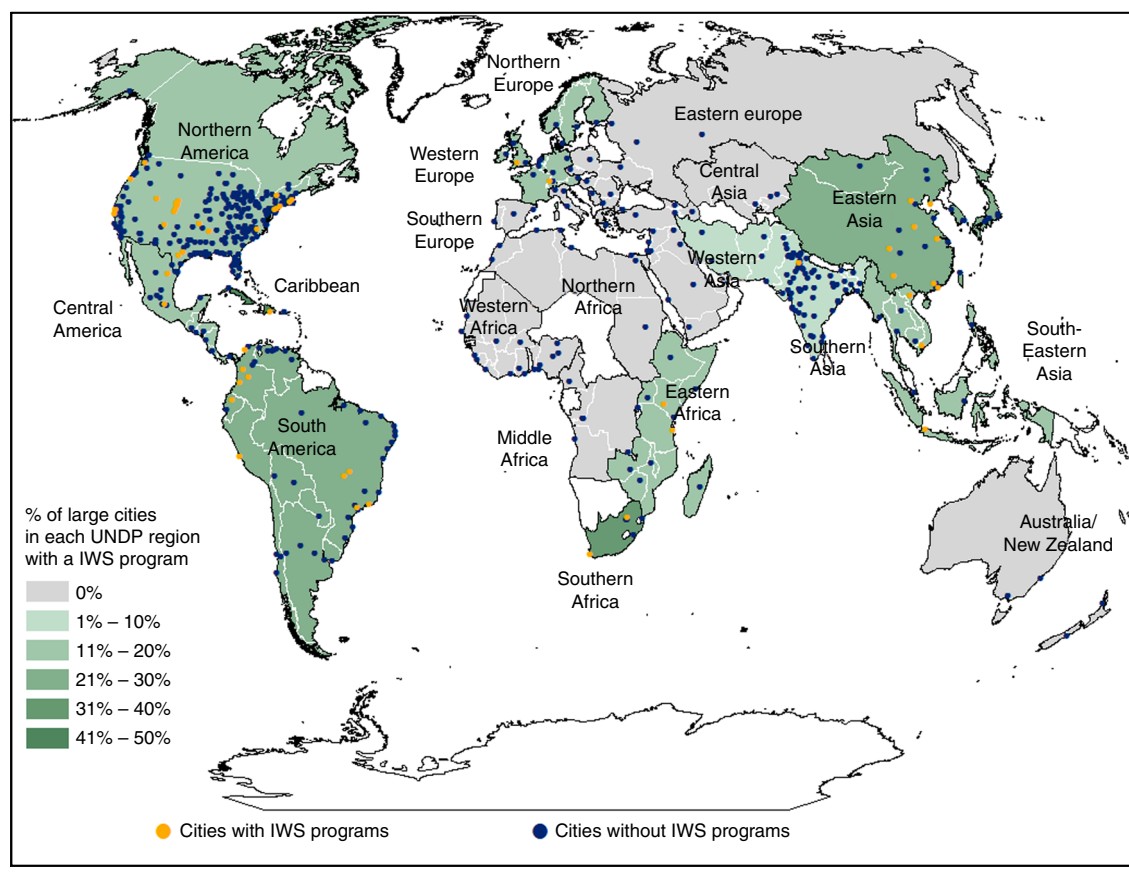

**Fig. 2** Percentage of large cities within each UNPD region that had an IWS program. Gray countries had no documented IWS programs and white countries had no cities large enough to qualify for inclusion in the CWM database. The remaining regions are shaded from the highest proportion of CWM cities with IWS (50%, dark green), to the lowest proportion of CWM cities with IWS (<10%, light green)

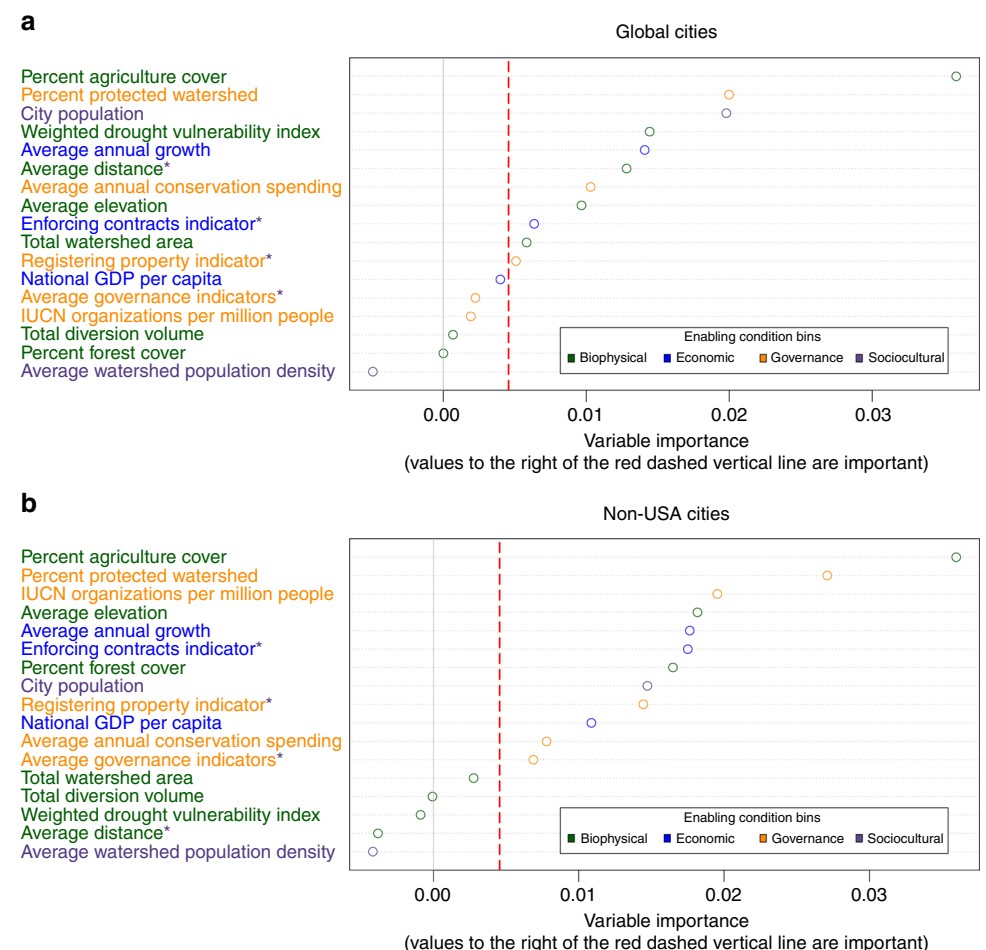

**Fig. 3** Relative ranked importance of enabling conditions variables. These are categorized in four main bins for **a** Global Cities, $n = 416$, and **b** Non-USA Cities, $n = 299$. Variables with an asterix could also be considered representative of enabling conditions in the sociocultural bin (see Supplementary Data 1 for relationship between representative data and enabling conditions). Variable Importance measures are a relative ranking of predictor variables, thus the absolute numbers on the X-axis do not have meaning outside of comparisons between predictor variable values. Values to the right of the red dashed vertical line are considered important in the model and those with higher variable importance values are more important than other variables with lower variable importance values

areas had both few large cities and few or no IWS programs (for example, Australia and New Zealand had 4 large cities, none of which had associated IWS programs identified by our research).

**Enabling conditions (variable importance).** We used random forest analyses to rank 17 predictor variables, each representing one or more key enabling conditions described in Fig. 1 (see Supplementary Data 1 for relationships between enabling conditions and representative data sets). The representative data were used to form two statistical models using different groups of cities: 1) The Global Cities Model containing all 416 large cities from the CWM; 2) The Non-USA Cities Model containing only the 299 large cities in the CWM that lie outside of the USA. We developed the second model due to the over representation of cities in the USA in CWM. These cities all receive the same value for country-level data, which would decrease the ability of these variables to explain variation in the data and result in artificially lower importance rankings. There were not enough large U.S. cities with IWS programs in our data set to justify a random forest model of only cities in the United States.

The two most important enabling conditions in both models were percent of watershed with agricultural land cover and

percent of watershed area designated as protected (Fig. 3). Both models also indicated that Average Annual Growth (the average annual growth rate of national GDP for 1994–2014) is important, possibly because economic growth may increase the resources available for payment for ecosystem services programs and rapid economic growth can increase impacts to water supplies from increased development without infrastructure and institutions in place to address these impacts[11]. Other enabling conditions, such as population (an indicator for number of potential stakeholders) and enforceability of contracts (a World Bank Indicator[14] used as proxy for both ability to enforce IWS agreements and secure land tenure) were relatively important in both models. Both models ranked water diversion volume and watershed population density as not important in predicting the presence of an IWS program relative to other variables.

**Differences in enabling conditions between groups of global cities.** Less variation in a variable would result in a lower importance ranking because it would be harder for the algorithm to differentiate outcomes. Anticipation of this relationship partially motivated our choice to build the Non-USA Cities model. For example, when applied to the Global Cities Model, the

**Table 1 Enabling conditions variables analyzed and their relationships with IWS**

| Count | Predictor Variable (using available data to represent enabling conditions) | Global Cities | Non-USA Cities | Enabling Condition (based on theory) | Enabling Condition Category |
|---|---|---|---|---|---|
| 1 | Average Annual Growth | ~ | + | Economic growth | Economic |
| 2 | Average Distance | − | | Resource location and arrangement | Biophysical |
| | | | | Proximity of actors to each other | Sociocultural |
| 3 | Average Elevation | − | − | Resource location and arrangement | Biophysical |
| 4 | Average Governance Indicators | | ~ | Strong existing institutions | Governance |
| 5 | City Population | − | − | Large/small number of actors | Sociocultural |
| 6 | Conservation Spending | − | − | Strong capacity among actors | Governance |
| 7 | Enforcing Contracts Indicator | ~ | ~ | Manageable transaction costs | Economic |
| | | | | Pre-existing market-based culture | Sociocultural |
| 8 | IUCN Organizations Per Million People | | + | Presence/absence of Intermediaries | Governance |
| | | | | Strong capacity among actors | Governance |
| | | | | Influential champion | Governance |
| 9 | National GDP per capita | | + | Economic growth | Economic |
| 10 | Percent Agriculture Cover | + | + | Clear threat or risk to ES provision | Biophysical |
| 11 | Percent Forest Cover | | − | Clear threat or risk to ES provision | Biophysical |
| 12 | Percent Protected Area | − | − | Secure land tenure and property type | Governance |
| 13 | Registering Property | + | + | Secure land tenure and property type | Governance |
| | | | | Pre-existing market-based culture | Sociocultural |
| 14 | Total Diversion Volume | | | Small resource area | Biophysical |
| 15 | Watershed Area | + | | Small resource area | Biophysical |
| 16 | Watershed Population Density | | | Large/small number of actors | Sociocultural |
| 17 | Weighted Drought Vulnerability Index | − | | Resource location and arrangement | Biophysical |

Variables are based on the biophysical, economic, governance and social-cultural enabling conditions and groups identified by Huber Stearns et al.[21] (Fig. 1). Not all variables identified by Huber-Stearns et al. were included in the analysis, because of unavailable or limited data (Supplementary Data 1). For each random forest model, important enabling conditions are provided a value (+/-/~) and unimportant conditions are left blank. Signs indicate the direction of the relationship between each condition and presence of IWS (Supplementary Fig. 1). Some important conditions have relationships that are neither positive nor negative overall, but vary in direction dependent on the underlying data gradients. These relationships are signified by ~ Some predictor variables were representative of multiple enabling conditions. In these cases, all potential representation are included in the Enabling Condition column

random forest algorithm identified other variables not based on national scale data as having greater predictive power (Fig. 3). Variable importance rankings differed in our models with the exception of the top two most important enabling conditions. We expected differences between the Global Cities and Non-USA Cities models because of the resolution of our data and the high number of USA cities in the data set. Several of the variables we included in our analyses were based on national scale data and were associated with the presence of IWS only in the Non-USA Cities model. Furthermore, governance and economic variables such as presence of IUCN organizations, World Bank aggregate governance indicators, and Gross Domestic Product (GDP), ranked relatively more important for predicting the presence of IWS in the Non-USA Cities model. In addition, the Global Cities model included a disproportionately high number of USA cities (117 of the 416 cities in the Global Cities Model were in the USA), all with the same values for socioeconomic and governance variables that were based on national-scale data.

Weighted Drought Vulnerability is ranked important for Global Cities and not important for Non-USA Cities, indicating that drought vulnerability is more associated with IWS programs in the USA than elsewhere. However, drought could be a driving factor to search for policy and program innovations only when other enabling conditions are already in place. Such interactions among variables potentially explain some differences between the important conditions found in our models. Enabling conditions may interact within each model, such that different values of one condition may impact the importance of other conditions. The following section describes additional analysis of the behavior of individual conditions within each model.

**Enabling condition directionality and behavior.** Enabling conditions are sorted into four general categories: biophysical, economic, governance, and sociocultural conditions (Fig. 1). While some enabling conditions were more important than others, we

found that in both models, important enabling conditions came from all four categories, with at least one important variable in each of the four categories (Table 1).

We used partial dependence plots for the random forest models to explore how individual enabling conditions could predict IWS programs across each range of values in the representative data (Supplementary Fig. 1). Partial dependence plots depict the relationship between an outcome and different values of predictor variables within a model, with all other predictor variables held constant. For many of the variables, the marginal effect on the outcome (presence or absence of an IWS program) was more pronounced at changes occurring between lower values (i.e. changes in watershed area at the lower range of area size). The partial dependence plots indicate that at low values, marginal changes in an enabling condition could have a large impact on predicting IWS, whereas at higher values, marginal changes did not increase an enabling condition's importance in predicting IWS. This may indicate possible thresholds of some predictor variables, above which increasing the variable does not influence the likelihood of IWS presence. Table 1 summarizes the direction of the relationship between each enabling condition and the outcome. For example, watersheds with higher percentage of area with agricultural land cover were more likely than otherwise comparable watersheds to contain a city with an IWS program. Watersheds with lower percentage of area protected were also more likely to contain a city with an IWS program.

**Expanding the scope of IWS.** Our results could be used in combination with local, context-specific data to guide decisions about sites for future IWS programs. We selected the top 5 enabling conditions from the Non-USA Cities model and divided cities outside of the US into top or bottom half of values for each enabling condition depending on the relationships described by partial dependence plots for each condition

(Supplementary Fig. 1). Using this ranking system, the following four cities most closely matched the top 5 enabling conditions associated with IWS programs, but do not currently have a program: (1) Dhaka, Bangladesh; (2) Guayaquil, Ecuador; (3) Dubai, United Arab Emirates; and (4) Leon, Mexico. An additional 37 cities met the characteristics for 4 of the top 5 enabling conditions (Supplementary Data 3). However, it would be critical to supplement an analysis of candidate cities with additional information about the places and people, as our results are not comprehensive of all required factors that enable IWS programs. For example, alternative approaches to managing urban water supplies such as desalination in Dubai could eliminate a need for IWS.

**Outlier cities**. Not all conditions must necessarily be in place for an IWS program to develop. Within our analysis, there are examples of cities with IWS that do not have all the identified enabling conditions in place. For example, Seattle in the US has an IWS program with 0% agriculture cover in its source watershed (97.59% forest cover) but a high percentage of protected area (82.1%). This situation reflects a history of land acquisitions by the Seattle Public Utilities, which now owns and protects a large portion of the watershed. In other countries such as Mexico, China, South Africa, and Colombia, we found additional cities with IWS programs even though important enabling conditions were not present. In some cases, as with Colombia, where four of their seven large cities employ IWS, there may be national level programs, legal instruments, or concerted NGO efforts to initiate and support IWS[9,11,26,27]. As hypothesized by previous research[22], while not all variables are needed for IWS to exist in a city, a combination of sufficient enabling conditions such as political support[28], strong conservation need[29], or outside conservation funding[30] could provide sufficient conditions for an IWS program to emerge. We emphasize that knowledge of which conditions are critical in specific contexts would be important for IWS program design, program success, and long term IWS sustainability.

Conversely, having important enabling conditions in place is not sufficient to ensure presence of an IWS program. Our database also contains examples of cities that do not have IWS programs even though they have high levels of enabling conditions, such as the candidate cities we identified. Having enabling conditions in places with no IWS program could indicate presence of a different management strategy that successfully achieves the same outcomes of protecting urban water supplies. For example, cities and countries could have alternative policies or management practices in place, from strong regulatory frameworks or more voluntary measures such as source water protection plans.

## Discussion

In our assessment of 416 cities with over 1.15 billion drinking water consumers, conditions representing a range of sociocultural, governance, biophysical, and economic factors were important for IWS presence. In comparing all major cities to only those outside the United States, two suites of important enabling conditions emerged in particular. We found that key enabling conditions for IWS programs in major global cities include the amount of watershed area in (1) agricultural land use and (2) protected designations. In general, threats or risks to ecosystem services can facilitate the development of IWS by increasing awareness of ecosystem service benefits and their need for conservation[22,25,29,31,32]. Places where ecosystem services have clear benefits to human communities are more likely to protect

ecosystem services, since beneficiaries have incentive to compensate service providers[25,28,30,33–35].

Greater percentage of agricultural land in the watersheds serving a city was an essential enabling condition for all of the cities in our sample and for the cities outside the USA (Fig. 3). Previous research suggests several mechanisms by which agricultural land can be an important factor associated with the development of programs such as IWS. Agricultural lands have long been a key area for the implementation of payments for ecosystem services type approaches, often due to large numbers of private landowners as ecosystem services suppliers, and the lack of specific regulation for concerns such as nonpoint source pollution[18,34,36]. Upstream agricultural land could further be associated with IWS programs and other environmental policy interventions because it can impact urban drinking water supply quality. Thus these land uses also present a ready opportunity for organized management actions[7,9,16]. Our results support the proposed linkages between agricultural lands, impacts on downstream water supplies, and existence of payment programs such as IWS, which have been cited as key drivers of IWS programs in cities such as New York in the US and Quito in Ecuador[15].

The percentage of protected area in source watersheds was the second highest ranking enabling condition in both models—although the relationship was negative. As the percentage of protected area increased, the probability a city had an IWS program decreased. IWS programs are designed to provide land owners with incentives to protect or enhance the watershed for the provision of water services[14]. Watersheds with large percentages of protected areas may not need further protection or incentives provided by IWS programs, so there is less motivation to develop programs in these locations. Additionally, source watersheds with a lot of land in the public domain may be easier to convert to protected status while watersheds with more private landowners or community-based tenure arrangements are a better target for IWS. In watersheds that have a low percentage of protected area, there may be increased opportunities for an IWS program as a way to influence management in the watershed and enhance water provision services via interventions on privately-managed land. Establishment of protected areas may also face additional hurdles in watersheds with large amounts of agricultural land[37], leading water managers to seek out market-based approaches such as IWS. Finally, a watershed with high percentage agricultural land and low percentage protected area could indicate increased risks to provisioning water services that have potential impacts to downstream water users.

Our research is one of the first attempts to quantitatively evaluate enabling conditions for IWS programs in cities across the world. Previous research on IWS has focused on individual or limited numbers of cases rather than global patterns. As a global-level analysis, our research begins to fill this gap by broadly testing factors associated with the existence of IWS programs. Our results should not, however, be interpreted as a mandatory or static checklist of all necessary factors to implement IWS or a similar policy. Though various conditions predict IWS presence, it is possible for IWS programs to emerge in a variety of situations. We identified the contextual conditions present in areas where IWS interventions already exist, and the conditions that were less relevant to the existence of IWS. Contextual details about the mechanisms underlying the emergence of IWS are important for understanding enabling conditions in specific places, even if the finer scale conditions vary. Conservation practitioners, in particular, could add from their experiences in the field to improve our understanding of what local conditions facilitate both the emergence and sustainability of IWS programs. For example, the lower cost of implementing IWS schemes as compared to other policy tools is a known factor in their

emergence[34,38]. Our analysis outside the US identified per capita GDP as positively associated with IWS presence, while conservation spending was negatively associated. The positive association with GDP could potentially suggest that a certain level of affluence is needed for IWS and that water providers are not able to spread the cost to users when users are predominantly poor. For conservation spending, it may mean that when spending is high, there are co-benefits to water quality coming from other investment actions that make IWS less necessary, similar to our interpretation for the negative relationship with the percentage of protected areas in the watershed. Where information on the cost effectiveness of various alternative strategies (including IWS) is available, it would enhance understanding of program emergence and sustainability.

Future research can further evaluate enabling conditions across a variety of contexts and scales to help establish clear relationships. Documenting and monitoring IWS performance are important for providing data to test the mechanisms through which enabling conditions are associated with the existence and sustainability of programs. Sub-national data and analyses would improve our ability to test enabling conditions and validate theory. Our results provide insights about general patterns and broad trends for large cities, but the nature of global synthesis can mask relationships between conditions that could explain variation within a region. Regional or country level analysis could provide more details about the mechanisms underlying how a program was operating (e.g. successfully or not) and the factors that are most important in initiating IWS programs, but our global-scale analysis is not designed or poised to take advantage of this higher resolution sub-national scale data. For example, funding sources, investors, and supporting organizations differ among IWS programs in Latin America[27], and within the US some major water utilities pay upstream landowners to change management practices (e.g. Denver Water, Colorado) or purchase land in the watershed (Seattle Cedar River Watershed).

Local conditions and context indeed matter for more nuanced analysis and local application of findings; however, understanding the general conditions that can make it more likely that a program will emerge is an important step in understanding where and how to dig deeper into finer grain analysis. Understanding the general conditions can provide partial explanation of program presence and evaluate the potential scope for expanding programs. Numerous researchers in this field have come to similar conclusions, in particular that the time is ripe to collect previously disparate lessons learned from case studies of ecosystem services and synthesize them for broader general conditions impacting presence of IWS (see synthesis provided in Huber-Stearns et al.). For example, Naeem et al.[39] call for the need to document initial baseline conditions, including the initial state of threats to services and important factors that will forecast service trends in the beginning of a program, and Ingram et al.[40] distill lessons learned about the use of ecosystem services, especially around understanding necessary institutional factors needed where governance may be weak. Recent research on PES programs more broadly identify key characteristics of buyers, sellers, and program specific metrics as key determinants of the spread and uptake of PES[13].

Implications and implementation of research on natural resource management is critical for practitioners. We have been working with collaborators at The Nature Conservancy (a non-governmental organization) on how to use the findings from this research to improve their IWS development program. When comparing potential locations for program investment, the most important conditions can be used to evaluate where IWS programs are likely present in comparable locations. In evaluating cities for program development, those that have similar characteristics to cities that do have a program may be good

candidates. We provide an example process in Supplemental Note 1 comparing Recife and Salvador, which are both coastal cities in Brazil using a ranking of cities based on the top 5 important conditions from our Non-USA cities model (Supplementary Data 2). Neither currently have an IWS program according to our research, although there are other cities in Brazil that do have a program. By comparing values for important variables delineated by this research, an IWS program is more likely present in Recife. This information is valuable when combined with local context and investment criteria to evaluate scope and expansion of IWS programs into new locations.

IWS programs emerge out of the interplay among numerous factors in complex social-ecological systems. What works in one place may not work in another because of the unique social and ecological contexts in each place. Our study takes an empirical approach in examining broad and globally available evidence on IWS programs and their enabling conditions. To elucidate particular conditions that enable innovative solutions in natural resource management, we emphasize that further cross-disciplinary and sub-regional investigations are needed.

## Methods

**Identifying cities with IWS programs**. The city water map: Our list of 534 global cities comes from the City Water Map, version 2.3[25] (CWM), a database by The Nature Conservancy containing information on large cities and their source watersheds. The original city list for CWM started with the World Urbanization Prospects (WUP) report conducted by the United Nations Population Division[1] that lists all current and previous world cities with a population > 300,000. Cities below this population threshold were added to the CWM from research on 225 cities with populations over 100,000 in the United States[2]. Data on the source watershed and specific withdrawal information was collected by searching water utilities directly, though in some cases no information was found. The final City Water Map list of cities contains 534 cities, including the world's 50 largest urban areas, the largest urban area in each country with > 750,000 people, and a representative sample of cities stratified by both geographic region and population range[1].

The CWM database contains a known bias resulting from data accessibility and availability that oversampled USA cities and undersampled Indian and Chinese cities[1]. The data were subset by removing all USA cities that met either of the following two criteria: (1) a population < 300,000, OR (2) no population data was available. This is based on the city population limit of 300,000 from the World Urbanization Projects report by UNPD[2] that the CWM database used to develop their database. Most of the cities in CWM under the 300,000 threshold were additions to the WUP report and creating this cutoff reduced the data set by almost 100 USA cities.

**Identifying cities with IWS programs**. Data on existing IWS programs were gathered from several sources. We analyzed the 416 cities that met the UNPD criteria for large cities (population > 300,000). We used Forest Trends' State of Watershed Investments bi-annual report[11] (29 cities identified) and a literature review of IWS programs to identify 59 cities in the CWM that have an IWS program using search engine Web of Science and publishing service ScienceDirect. A search was conducted for title, abstract, and keywords only using the search terms "payment* for ecosystem services" OR "payment* for environmental services" OR "payment* for water* services" AND "water*". Web of Science results listed 136 articles and ScienceDirect returned 91, which, excluding duplicated articles, produced a library of 171 articles.

Much of the program information was collected from the State of Watershed Payments annual report, produced by Forest Trends[11]. The Forest Trends report and literature search were reviewed for IWS programs that met two criteria; (1) they provide water for a city in the CWM database, and (2) drinking water protection is specified as a program goal. The list of cities that have met the IWS criteria include those with Demonstration Projects that are focused on drinking water because they are actively managing drinking water using a IWS program. For this research, cities with IWS programs ($City_{IWS}$) are those cities within the CWM database with a IWS program identified by either the Forest Trends report and/or the literature review (Supplementary Fig. 2). Cities with no known IWS program are denoted by "$City_{no\ IWS}$." Of these 59 cities with IWS, 53 met the UNPD criteria for large cities. We defined IWS as transactional arrangements (in cash or in-kind) between two or more parties that compensate a land manager for protecting drinking water supplies for urban beneficiaries[11,22]. Our list of enabling conditions built on a synthesis of theory and case studies on payments for ecosystem services conducted by coauthors on this paper[22]. We identified global data sets for the variable (e.g. city population, watershed area) or, when necessary, for a proxy indicator that represented the variable (e.g. Property Rights Index represents land

ownership and access). We intentionally targeted data for all four condition categories (biophysical, economic, governance, and sociocultural data) identified by Huber-Stearns et al. in an attempt to represent as many different types of potentially important characteristics as possible. All city data is available in Supplementary Data 3.

### Enabling conditions concept and data

*Enabling conditions concept.* The original concept and list of enabling conditions is derived from previously published work[22]. Enabling conditions are defined as factors that increase the likelihood of an intended change in the governance approach, strategy, or management regime. Enabling conditions, by definition, facilitate the emergence or sustainability of a particular environmental policy, while the absence of key enabling conditions can present a barrier to management or sustained policy action. In this initial publication we summarized existing literature on the concept of enabling conditions and synthesized the information into a list of potential conditions, grouped by category (Fig. 1). Although these categories provided more structure for the presentation of conditions, it is important to note that the conditions in each theme were identified from a variety of disciplinary perspectives and fields, journal types, and author considerations, so no one theme was solely represented by one discipline.

**Enabling conditions data**. Here we distinguish between EC variables, those broad conditions identified by Huber-Stearns et al.[22] and Representative Data, the actual data used in the analysis. Information from 14 data sets were collected, processed, and integrated into a relational database (See Supplemental Data 1 for relationship between EC variables and representative data sets). For this study we targeted global data sets to emphasize standard measurements for each indicator. For some EC variables no representative data was available with global coverage. In some cases representative data could potentially represent multiple EC variables (Supplementary Data 1). For example, the number of IUCN organizations per million people could represent the presence of an influential supporter of PES such as a politician or prominent NGO, the presence of strong intermediaries, and strong capacity among actors. It these cases it is also possible the representative data reflects a combination or interaction of EC variables.

**Statistical analysis**. Statistical analyses were performed in R version 3.2.3[41] with some pre-processing of geospatial data in ArcGIS[42] within an equal-area Mollweide projection[43].

**Water supply origin and water source characteristics**. The origin of the water supply for each city and characteristics of the watersheds were described using CWM diversion type categories and volumes, combined with delineations of the surface and groundwater basins that serve each city[25,44,45]. Percent ground or surface water was categorized in one of six types: primarily surface water ( > 75% of diversion volume from surface sources); mixed sources (50–75% surface volume, 25–50% surface volume, or 1–25% surface volume); groundwater sources only, or no available data. Surface Water includes all diversion types except groundwater and alluvial aquifers. Watershed area was calculated as the combined area (km[2]) of all watersheds and groundwater basins being used for drinking water for each city. Percentage of protected area is from IUCN-designated protected lands within this total area[46]. Land cover types (percentage forest and percentage agricultural and/or pastoral) were calculated for the source watersheds and basins[44] of each city and grouped based on classification per Supplementary Table 2. For cities with mixed above and below-ground water sources with diversion volumes available for each, land cover was weighted by diversion volume. If diversion volumes were not available for all sources, land cover was represented by the sources with available data.

**Calculating post-stratification weights**. Post-stratification weights were calculated for each city in the CWM to further address sampling bias and adjust the distribution of cities to reflect real city distributions[45]. Using the World Urbanization Projects Report (WUP)[1] the proportion of cities within each geographical region was calculated for each of 5 city population classes1 (Supplementary Table 3). Region was used as opposed to country because some countries have few or no cities in the CWM data set. The WUP report originally supplied the base data for the CWM and the geographical regions and population classes are described in the report as well. Proportions of each city class were calculated and used to determine a weight field (# database cities in region class/sum of UNPD cities per region) that adjusts city data proportions to the WUP report proportions.

**Variable selection**. Thirty candidate variables from existing data sets were identified to represent potential enabling conditions as identified in Huber-Stearns et al.[5] Variables either directly quantify conditions, as in the case of biophysical and economic characteristics, or serve as recognized proxies of city characteristics. Predictably, many of these variables are correlated, as they are based on shared information (i.e., several of the country level economic indices are calculated using GDP). Collinear and replicated variables were excluded. Selection was based on analysis of spearman pairwise correlations and variance inflation factors[47]. The R

package Corrgram v1.10[48] was used to calculate correlation coefficients. Of the 30 variables tested, 18 were found to be correlated with at least 1 other variable at corr > 0.7, indicating high collinearity[17]. Supplementary Table 4 provides the correlation coefficients between highly correlated variables (corr > 0.7) and justification for which of the correlated variables were selected for inclusion in the final models. In addition to the spearman correlation coefficient, variance inflation factor (VIF) was calculated using R package car[48] using the full database as well as a subset of the cities contained only non-USA cities, though not all variables could be included because of missing values. VIF is calculated as 1/(1-R2) from a linear model and estimates how much the variance of a coefficient is inflated from linear dependence with other predictors. A higher VIF value indicates that the variance (the square of the standard error) is larger than if the predictor were not correlated with other predictors. VIF were calculated iteratively by sequentially dropping the predictor with the largest VIF, recalculating with the remaining variables, and repeating until returned values were under the preselected VIF threshold of 3[49]. Supplementary Table 5 provides the VIF values for our final list of 17 variables, with any values exceeding our threshold of 3 in bold.

Three representative data sets (Conservation Spending, Average World Bank Governance Indicators, and National GDP per capita) did not meet the VIF criteria, but were included in the model analysis because there were no other proxy variables for the EC variable they represented. After reducing both the number of cities and the representative data, the final database used for analysis contained 416 cities and 17 variables, representing 14 of the EC variables described by Huber-Stearns et al. (Correlation coefficients provided in Supplementary Table 6). A final data table with all representative data is provided as Supplementary Data 3.

**Random forest model**. We determined the predictive importance value of our representative data using a random forest model of classification trees[23]. This model was selected because inference trees are robust when regressing data with high dimensionality, which is a situation with many predictor variables compared with the number of data points[50], often referred to as a large p, small n problem. Previously published research on enabling conditions for IWS programs often discuss only one or few enabling conditions, but our analysis allowed us to build a model using interactions between variables as opposed to evaluating fit to an existing model or assumptions. Using machine learning to consider many variables at once allows us to rank those variables in terms of importance for predicting the presence or absence of IWS programs. Logistic regression was considered as a potential model, but initially resulted in perfect separation, likely due to the small minority class and high dimensionality characteristics of the data. The random forest approach has been widely used in the medical field for situations with highly unbalanced data with varied and potentially interacting predictor variables[50,65,66], and is becoming more prevalent in the conservation and natural resource management literature, especially when attempting to evaluate global patterns[67–70]. Random forest methods also reduce issues of bias toward the majority class that can occur with unbalanced data sets in logistic regression[71,72], important because in this data set cities with IWS programs represent the minority class.

Random forest models are a type of machine learning algorithm that consist of many individual decision trees constructed with random subsets of predictor and dependent variables. Each tree in the random forest model predicts the presence or absence of a IWS program for a CWM city using a random subset of data and predictor variables. The model ranks all variables according to aggregate prediction performance in the forest of individual trees[23]. The model constructed by the random forest classification technique allows us to rank variables in terms of importance in predicting the presence or absence of IWS in a given city. We selected this classification system specifically for high classification accuracy and the ability to model complex interactions between predictor variables[24]. We used the R package Party[51] because its functionality is particularly well suited for unbalanced data sets with high dimensionality[50], can address missing data[52], and has the capacity to reduce bias from predictor variable type and correlated predictors[25,44,45].

The data were split 80/20 (pareto principle) for training and test sets and not transformed. We weighted enabling conditions data to represent city distribution regionally and globally using UN statistical region boundaries (described above in the section titled Post stratification weights). Given the unbalanced nature of the dependent variable (IWS presence or absence), several strategies were attempted to address potential bias in the model due to the small size of the minority class (only 11.5% of modeled cities contain an IWS program because some cities were not included in the models). To address this class imbalance, the data were adjusted four different ways before modeling: (1) the larger class (cities with no IWS programs) was undersampled[53], (2) the smaller class (cities with a IWS program) was upsampled, and (3) new minority class were created using the Synthetic Minority Over-sampling Technique (SMOTE) function[54] in R package DMwR[55], and (4) weights were incorporated in the random forest classifier which made the classifier cost sensitive and penalized the model fit for misclassifying the minority class. The final models reported here addressed class imbalance by incorporating a weight class in the random forest model as this method produced models with higher predictive ability (predictive performance described below) better than other options.

Each model contained 8,000 trees[56], with the number of preselected variables (mtry) set to 4 (calculated as the square root of the number of predictors[57]) and all

other parameters set to default. We used the unbiased permutational variable importance measure (function varimpAUC), because it is particularly suited for unbalanced response classes[58]. The varimpAUC output is also a non-conditional variable importance measure that can be computed with missing data[52]. With this approach correlations between predictor variables need to be addressed separately, which was done using a spearman correlation test as described in the Variable selection section above. Predictor ranking was evaluated using mean decrease accuracy because of varying scales of measurement, and correlation among predictors[59]. Predictive power of each model was evaluated using a standard metric, area under the Receiver Operating Characteristic curve (AUC), which assesses classification accuracy[56]. Values for AUC range from 0.5 to 1 and the closer to unity, the more accurate a model, where models with a value of 0.7 are considered reasonable and those with values > 0.8 considered strong[60]. Both models presented here performed at AUC values of > 0.7. The relationships between individual predictors and outcome (presence of IWS) was evaluated using partial dependence plots via R package mlr[61]. Partial dependence plots reveal the relationship of individual conditions within each random forest model by integrating out (and thus controlling for) other factors. Greater $y$ values indicate that an observation for a specific variable is associated with higher probability for classifying a city as having an IWS program. These plots depict the marginal effect of the variable to provide an average trend of individual variables within a model by integrating out all other variables[62,63].

**Code availability**. All data and code are available on GitHub (https://doi.org/10.5281/zenodo.1403842) including the code used for the shiny app reference in the data availability section. Link: https://github.com/cromulo/IWS.

## Data availability

All data used in this research is open source. The data sets used for representative data of the enabling conditions are freely available and citation information can be found in Supplementary Data 1. The full list of enabling conditions (including those not tested), and the specific values used in our analysis are provided in Supplementary Data 3. City and associated watershed information was obtained from The Nature Conservancy City Water Map[36] with permission. IWS program information was collected from a combination of (1) Forest Trends' Ecosystem Marketplace State of Watershed Investment survey[11] with permission and (2) literature review detailed in the Methods. The data sets and descriptive statistics can be accessed from this shiny app site as well: https://cromulo.shinyapps.io/InvestmentInWatershedServices/.

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

## Acknowledgements

We thank D. Gill, C. Kennedy, G. Bennett, and D. Shemie for reviewing an earlier version of this manuscript, and J. Goldstein, G. Bennett, T. Gartner, and N. Carroll for guidance on early ideas in this research. We also thank K. Wallen for a pre-submission review and assistance with figures. We are grateful to Forest Trends Ecosystem Marketplace for the use of their State of Watershed Investment data for this analysis. This work was supported by the National Socio-Environmental Synthesis Center (SESYNC) under funding received from the National Science Foundation DBI-1052875. The authors participated in the Enabling payments for watershed services 2015–2016 SESYNC project (https://www.sesync.org/project/graduate-student-pursuit-rfp/enabling-payments-for-watershed-services).

## Author Contribution

C.L.R., D.E.B., R.C.R., R.M. and S.P. conceived the original project; all authors contributed to study design and collected data; C.L.R., J.H-F., S.J.M.C. and S.P. performed analyses and wrote methods; J.H-F., S.C., C.L.R., D.E.B., H.H-S., R.C.R. and S.P. wrote the Introduction, Results and Discussion. All authors discussed results and interpretation, as well as reviewed and edited the manuscript at all stages.

## Additional information

**Competing interests:** The authors declare no competing interests.

