## [Peer Review File · Nature Communications]

Reviewer #1 (Remarks to the Author):

Review

This paper aims to provide insight in the enabling conditions for payments for watershed services in large urban areas with more than 300,000 inhabitants. It tries to do so by using a random forest model of classification trees, applied to The Nature Conservancy's City Water Map database and other global databases containing information about 17 indicators, varying from watershed area to city population. The main outcome and conclusion of the paper is that payment schemes are more likely to be present in watersheds with more agricultural land and less protected areas. These results seem somewhat obvious: the higher the threat (agricultural area), the more likely action will be taken to protect urban water sources, while designated protected areas are simply an alternative policy instrument, reducing the need for additional action such as a payment scheme for watershed services.

Although the topic is interesting, the contribution of this study to the existing literature is questionable. Only 13 percent of the 416 cities (n=53) included in the analysis actually implement payment schemes. No control is included, for example, for the age of the payment schemes or their institutional-economic design. The title of the paper does not reflect its content. Often, payments for watershed services serve multiple goals, not only environmental ones (e.g. Bulte et al., 2008; Brouwer et al., 2011). No control is included for this either. It is also unclear how cities were matched, i.e. made comparable, in order to isolate the impact of the 17 control variables included in the statistical analysis. Almost a quarter of these 17 control variables furthermore seem to have switching signs (+/-), including the share of protected area in the non-USA cities model (Table 1).

What is more of interest in this field is the evaluation of the incremental impacts of payment schemes for ecosystem services compared to other policy instruments, such as regulation (see the 2015 publication by Naeem et al. in Science about this), or the cost-effectiveness of this instrument compared to traditional engineering solutions to secure safe drinking water supply. This paper does not look at this ("Here we focus on the emergence of IWS programs for cities rather than an evaluation of their impacts"), whereas cost-effectiveness is expected to be one of the key driving forces behind the implementation of this policy instrument. Their emergence as a novel market based instrument has been described in various papers, special issues in journals and textbooks about ecosystem services since 2000. User financed schemes which reduce reliance on limited government budgets for environmental protection and conservation have been propagated, especially in the developing world, as an important impetus behind their emergence (e.g. Wunder, 2015; Engel et al., 2008). Interestingly in this study, however, GDP per capita has a positive effect on their emergence (higher income countries applying payment schemes more likely, in the non-USA model), while conservation spending has a negative effect (higher conservation budgets reduce the need for payment schemes, in both models). No further discussion or explanation is given, nor is there any link provided to the literature in this field.

One could argue that the value added of this study is that it tries to quantitatively identify the significance of the role of various driving factors. The question is whether the high level of analysis and the chosen global variables are able to capture the relevant socio-economic, socio-cultural and governance dimensions. I'm not familiar with the applied statistical model, but it would have been good to test the sensitivity of the results to the applied statistical model. As a model robustness check, a simple probit or logit model could have been estimated, for example, regressing the variables representing enabling conditions on the binary variable identifying whether a city has a payment scheme in place.

In conclusion, I don't believe this paper adds enough to the existing literature to warrant publication in Nature. The topic is interesting and the quantitative approach commendable, but the model is too high level and the comparability of the cities and 53 schemes across the implementing and non-implementing cities questionable. More interesting is the question what the impacts are of existing payment schemes on environmental and socio-economic conditions and to what extent these impacts are driven by enabling conditions, including their institutional design, and how cost-effective they are in inducing the necessary behavioral change in the linked urban-agricultural landscape.

References

- Brouwer, R., Tesfaye, A. and Pauw, P. (2011). Meta-analysis of institutional-economic factors explaining the environmental performance of payments for watershed services. *Environmental Conservation*, 38(4): 380-392.
- Bulte, E.H., Lipper, L., Stringer, R. and Zilberman, D. (2008) Payments for ecosystem services and poverty reduction: concepts, issues and empirical perspective. *Environment and Development Economics* 13: 245–254.
- Engel, S., Pagiola, S. and Wunder, S. (2008) Designing payments for environmental services in theory and practice: an overview of the issues. *Ecological Economics* 65(4): 663–674.
- Ezzine-de-Blas, D., Wunder, S., Ruiz-Perez, M., Moreno-Sanchez, R. (2016). Global patterns in the implementation for environmental services. *PLoS ONE* 11(3): e0149847.
- Naeem, S., Ingram, J., Varga, A., Agardy, T., Barten, P., Bennett, G. et al. (2015). Get the science right when paying for nature's services. *Science*. 347 (6227).
- Wunder, S. (2015). Revisiting the concept of Payments for Environmental Services. *Ecological Economics* 117: 234-243.

Reviewer #2 (Remarks to the Author):

The manuscript "Global State and Potential Scope of Investments in Watershed Services for Large Cities" builds an impressive database of cities around the world to better understand what conditions lead to implementation of investment in watershed service (IWS) programs. It uses a random forests classification algorithm to parse out which variables in their dataset are most associated with IWS programs. The manuscript is well structured and is an interesting approach to look for general associations of where IWS programs have sprung up. To my knowledge no other work has compiled such a dataset nor attempted this kind of an analysis.

There are areas where the paper could be improved or clarified. One theme throughout my comments is whether the authors can say more about contextual conditions, or how some factors may be important conditional on other factors, which is what we might expect when implementing policy in complex social-ecological systems. The authors present only the globally significant results which, in my mind, mask what could be a much richer exploration of what conditions matter when, and for whom. Without providing some examination of how context matters, the main message falls a bit flat to me.

Below are what I see as general issues, followed by some more specific comments. I also attached pdf and word documents with some comments (sometimes redundant with the comments below) for further consideration.

GENERAL COMMENTS

1. The overall approach taken by the authors is one that assumes there are generalizable conditions that enable or encourage IWS programs to start in a given location. The authors state that conditions associated with IWS programs "have not been analyzed in aggregate or evaluated for relative importance" (line 53) and that we "need to evaluate the factors associated with IWS programs across many contexts" (line 55). However, why would we expect a consistent set of conditions to be enabling across such a variety of different contexts? Research focusing on program or policy implementation in social-ecological systems shows just how much context and local conditions matter for successful outcomes (e.g., Leslie et al. 2015, Alexander et al 2016, etc.). It would be helpful for this approach to be justified.

2. Relatedly, it seems there may be more information that the authors could present related to the interactions of contextual conditions. In the methods the authors say they "build a model using interactions between variables" (line 304) but these interactions are not presented or discussed in the paper as far as I can tell. How do interactions matter? What do they reveal about potential factors that matter conditional on other factors?

3. The authors use random forests algorithm to analyze the dataset on urban IWS programs. However, little justification is provided for why this method was chosen. What are strengths and

weaknesses of this versus other methods like multiple linear regression (e.g., Oliveira et al. 2012, Smith et al. 2013) or maybe even qualitative comparative analysis (QCA) (Ragin 2007)?

4. Admittedly, I have only casual knowledge of random forest algorithms, so my perspective on methods is one of a slightly informed but somewhat lay reader. In this regard, I am used to seeing decision trees as an outcome of classification methods like random forests, which often at least helps show how splits in the data reveal different conditional patterns and processes, which may relate to different contexts and provide some evidence that the authors call for in the discussion (lines 241-252). However, in this manuscript (throughout the main text and the supplementary materials) the authors rely only on the variable importance diagram (Figure 3) as the main outcome from which they rest all their results and interpretation. Why have the authors chosen to rely on such a limited set of results?

5. The authors rightly note that their analysis would need to be supplemented with other information to inform actual decision-making. But the limitations to this "technocratic" approach could be better highlighted. Namely, that it does not take into account local economic (what are the costs and benefits of a program in a given place) (Zheng et al. 2013), political, or social conditions that may affect the appropriateness of a program.

EDITORIAL COMMENTS

line 66

The title of this section is "Global and Regional Distribution of IWS" which implies the sample is representative and that this is, indeed, indicative of the distribution of IWS. However, no information is given as to whether the "stratified sample" (line 68) developed by McDonald et al was random or representative, nor the sampling procedure for the other 59 cities the authors "identified" (line 69). Were these randomly sampled? Does the reader have any assurance these are indeed representative of global distributions? If not, its hard to reliably interpret any of the regional breakdowns that are presented.

lines 84, 90

the use of the word "dataset" is confusing here, as it seems in the text the authors are referring to just one variable from a single dataset.

line 102

suggest replacing “not important” with “not statistically significant at a 0.05 level of confidence.” This is not the same as importance.

line 121

how did breaking the data into two models confirm the authors' “understanding of how the random forest algorithms identify important predictors”? What confusion did the author's have without this confirmation?

line 137

To the reader its unclear how the partial dependence plots show how the variable relates to whether there is or is not an IWS program. This is a binary condition, but the plots are continuous. I think the authors mean something like: These plots indicate the probability of that variable being present in cases where an IWS program is present relative to cases without an IWS program.

line 139

Does “the outcome” refer to the presence or absense of an IWS program?

line 140-141

how much confidence can we have in such statements? Can the authors include confidence intervals around these probabilities in the partial dependence plots?

line 319

Supplementary Table 3 seems to be missing

Table 1

some of the variables (e.g., “IUCN Organizations Per Million People” and “Registering property”) are repeated across multiple conditions. Does this mean the variable is repeatedly included or just that this single variable represents several different governing conditions? This is potentially confusing to the reader.

Figure 3

Indicate the meaning of the red dotted line in the table notes or on the figure directly.

Supplementary Figure 1

* Label x axis in these plots. Is this the range of the data for a given variable? The y axis is probability of what?

* The Figure caption refers to Figure 3. There are no lines in figure 3 of the main text? Or do the authors mean this figure (Supp Fig 1)?

* The Figure caption says "Line shapes in Figure 3 explain directionality of prediction ability and thresholds, but scale and numbers do not represent accuracy or prediction". It is not clear to me whether this is supposed to refer to Figure 3 or Supplementary Figure 1, nor is it clear to me what this phrase means.

* The Figure caption notes the variables are displayed in rank order. Indicate whether this is left to right or top to bottom.

Supplementary Table 2

Formatting — the table is split into two pages, I assume the authors intended this to be in landscape format.

Supplementary Methods (SM)

Sometimes the list of figures and tables here relative to the main text is confusing. I recommend that all figures and tables in the SM document be listed as SM Table 1, SM Table 2, etc. to avoid confusion.

WORKS REFERENCED

Alexander, S.M., Andrachuk, M. & Armitage, D. (2016). Navigating governance networks for community-based conservation. *Front. Ecol. Environ.*, 14, 155–164.

Leslie, H.M., Basurto, X., Nenadovic, M., Sievanen, L., Cavanaugh, K.C., Cota-Nieto, J.J., Erisman, B.E., Finkbeiner, E., Hinojosa-Arango, G., Moreno-Báez, M., Nagavarapu, S., Reddy, S.M.W., Sánchez-Rodríguez, A., Siegel, K., Ulibarria-Valenzuela, J.J., Weaver, A.H. & Aburto-Oropeza, O. (2015). Operationalizing the social-ecological systems framework to assess sustainability. *Proc. Natl. Acad. Sci.*, 201414640.

Oliveira, S., Oehler, F., San-Miguel-Ayán, J., Camia, A. & Pereira, J.M.C. (2012). Modeling spatial patterns of fire occurrence in Mediterranean Europe using Multiple Regression and Random Forest. *For. Ecol. Manage.*, 275, 117–129.

Ragin, C.C. (2007). *Qualitative Comparative Analysis Using Fuzzy Sets (fsQCA)*. In: *Config. Comp. Anal.* (eds. Rihoux, B. & Ragin, C.). Sage Publications.

Smith, P.F., Ganesh, S. & Liu, P. (2013). A comparison of random forest regression and multiple linear regression for prediction in neuroscience. *J. Neurosci. Methods*, 220, 85–91.

Zheng, H., Robinson, B.E., Liang, Y., Polasky, S., Ma, D.-C., Wang, F.-C., Ruckelshaus, M., Ouyang, Z. & Daily, G.C. (2013). Benefits, costs, and livelihood implications of a regional payment for ecosystem service program. *Proc. Natl. Acad. Sci. U. S. A.*, 110, 16681–6.

Reviewer #3 (Remarks to the Author):

Review of Global State and Potential Scope of Investments in Watershed Services for Large Cities

The paper offers an interesting quantitative analysis of enabling conditions for watershed conservation schemes for large cities. It is of critical importance to gain a better understanding of the factors important for emergence of water services protection schemes on the ground and the research field has gained momentum in recent years. This is evident from the paper and the authors demonstrate a good overview of the current state of knowledge. The paper is a global analysis, which is the first of its kind according to the authors. Other studies in the literature have tended to focus on individual case studies or a few cases. I agree with the authors that the global scale analysis of payment schemes for ecosystem services is rare in the literature and the paper offers a novel contribution. The paper uses a decision tree approach to the analysis of enabling factors. The method choice appear suitable and is also

The paper is well written and clearly organized and appears to follow the journal style and format requirements. I have however some suggestions for improvement of the paper both in terms of clarity and in terms of justification of the approach taken.

Major comments:

1) The caption for Figure 1 states the figure is modified from Huber-Stearns et al. It looks like a copy – at least I can't tell what the modifications are. From the methods section it is clear that there is some overlap between the authors of this paper and the current paper, so that the authors are only copying from themselves. Even so, it is not appropriate to claim that the current paper offers more novel material than it actually does.

2) The paper refers to IWS as the core concept. It is not clear to me how this concept differs from PES and why you choose to change the terms used in the literature.

3) Table 1: IUCN organisations per million people appears as indicator of three different enablers. While I appreciate that it is difficult to identify very accurate indicators of all the variables indicated in Figure 1 it is misleading that include all the variables in Table 1 when three of them are the same variable. I think it would be better to reduce the number of variables in the table and explain in the text that some of the variables do not clearly represent only one of the types of enablers. This is also the approach the authors take in Figure 3.

4) Methods:

The authors say that models are build based on 17 datasets – I interpret this as the different enablers. Is his correct?

I get the impression that the difference in the models developed is due to the high correlation between variables. However, I am not entirely sure that this is what happening. What do you use the correlation analysis for? Could you clarify?

It is also not clear how you choose the preferred model. Is this based on the AUC ? Is this part of the Random Forest approach or do you select the preferred model?

Furthermore, while the random forest approach appears interesting it is not clear what is gained relative to more simple regression type models predicting the presence of IWS. The authors argue that the random forest approach allows the researcher to capture complex relationships between enabling factors. However, it is not really clear how this strengthen the analysis and findings in the study presented. The supplementary material indicates that the classification is a simple tree with threshold cut-off points to determine probability of a city having a IWS. I have probably misunderstood this, but I think the paper would benefit from a clearer description of how the more complex methods enrich the analysis.

5) I like Figure 3 but it is not clear to me how to interpret the values on the x-axis. Could you explain?

6) It is not clear to me what you use the partial plots for. The authors refer to a Figure 3 in the supplementary material that would probably help but the Figure does not appear to be submitted?

Supplementary material:

The supplementary Tabel 1 gives the description of how the enabling factors have been operationalized in the quantitative analysis by outlining the variables chosen and the data sources. This is very helpful and gives a clear overview of how the authors have ended up with due to data shortage at the global scale.

Supplementary material table 2:

Format of the supplementary material – Table 2 – needs reformatting, as it is not possible to see which cities link to some of the enablers.

The caption in table 2 refers to partial plots in supplementary material Figure 1. This Figure does not appear to be in material ?

Minor comments:

Line 27: Not clear what you mean by water risk.

Line 159: A double full stop.

Line 193: A double full stop.

RESPONSE TO REVIEWERS

Re: manuscript titled *Global State and Potential Scope of Investments in Watershed Services for Large Cities*

We thank the reviewers for their careful consideration and supportive comments. We feel that the reviews were insightful and have helped strengthen our revised manuscript. In this letter we detail how we have addressed all comments through general changes, and specific changes described in the table beginning on the following page. Reviewer comments are provided in *italicized text*. Throughout, line numbers in referee comments refer to the original manuscript and line numbers in responses refer to the revised manuscript. We have numbered comments to indicate both the reviewer and comment number (1.X indicating all comments from Reviewer #1).

General Changes

In addition to changes regarding specific comments, we have also added explanation to the Introduction regarding the Random Forest modeling algorithm to address reviewer's critical questions regarding the use of the model. The additional model discussion in the Introduction provides readers with important context before continuing to results and discussion. Initial attempts at logistic regression resulted in perfect separation and we elected to use a different approach. We have included text regarding preliminary logit regression modeling in Lines 66-67, 347-349, and 358-36 and explanation of why this particular analysis did not fit our needs. The random forest model was selected because our data and research question present a similar structural pattern as that of research questions addressed using random forest in other fields. For example, random forest has been widely used in the medical field for situations with highly unbalanced data with varied and potentially interacting predictor variables (Boulesteix et al. 2012; Qi et al. 2005; Strobl et al. 2007). This model is becoming more prevalent in the conservation and natural resource management literature, especially when attempting to evaluate global patterns (Gutierrez et al. 2011; Edgar et al. 2014; Gill et al. 2017). Given the availability of the data, we selected an analytical approach that identified the predictor variables that best explained the outcome variable of presence/absence of IWS programs. Using random forest allowed us to account for higher-order interactions, nonlinear relationships among predictors, and 'small n / large p' models per Gill et al. 2017. Additionally, in order to clarify how to use our analysis, we have added Box 1 with information regarding our collaboration with The Nature Conservancy and how they are applying our results to their Water Funds program.

Several comments from reviewers revolved around a critique of the variables considered in our analysis. Some comments suggested other variables for inclusions, such as contextual conditions,

or those factors that are conditional on others. We agree with the reviewers and have noted that such variables are critical to consider for implementing policy in complex social-ecological systems. We have modified our manuscript to bring this important context forward for readers. The findings by Huber-Stearns et al. (2017) offers a qualitative review of current theories regarding potentially important variables, but the analysis of our study provides a quantitative evaluation of relative variable importance.

Boulesteix, A.L., Janitza, S., Kruppa, J. and König, I.R., 2012. Overview of random forest methodology and practical guidance with emphasis on computational biology and bioinformatics. *Wiley Interdisciplinary Reviews: Data Mining and Knowledge Discovery*, 2(6), pp.493-507.

Edgar, G.J., Stuart-Smith, R.D., Willis, T.J., Kininmonth, S., Baker, S.C., Banks, S., Barrett, N.S., Becerro, M.A., Bernard, A.T., Berkhout, J. and Buxton, C.D., 2014. Global conservation outcomes depend on marine protected areas with five key features. *Nature*, 506(7487), pp.216-220.

Gill, D.A., Mascia, M.B., Ahmadi, G.N., Glew, L., Lester, S.E., Barnes, M., Craigie, I., Darling, E.S., Free, C.M., Geldmann, J. and Holst, S., 2017. Capacity shortfalls hinder the performance of marine protected areas globally. *Nature*, 543(7647), pp.665-9.

Gutiérrez, N.L., Hilborn, R. and Defeo, O., 2011. Leadership, social capital and incentives promote successful fisheries. *Nature*, 470(7334), p.386.

Qi, Y., Klein-Seetharaman, J. and Bar-Joseph, Z., 2005. Random forest similarity for protein-protein interaction prediction from multiple sources. In *Pacific Symposium on Biocomputing*. Pacific Symposium on Biocomputing (pp. 531-542).

Strobl, C., Boulesteix, A.L., Zeileis, A. and Hothorn, T., 2007. Bias in random forest variable importance measures: Illustrations, sources and a solution. *BMC bioinformatics*, 8(1), p.25.

Reviewer #1 (Remarks to Author)

1.1: These results seem somewhat obvious: the higher the threat (agricultural area), the more likely action will be taken to protect urban water sources, while designated protected areas are

simply an alternative policy instrument, reducing the need for additional action such as a payment scheme for watershed services.

While our results certainly are consistent with anecdotal evidence, this is the first study to look at a broad sample of cities globally and provide quantitative evidence that source watersheds with more agricultural areas are more likely to take steps to protect water supplies. In a previous study, Huber-Stearns et al. (2017) aggregated anecdotal evidence regarding potential enabling conditions for PES programs. Our study provides a quantitative evaluation of the relative importance of these enabling conditions. We are glad to see that the findings of our research makes sense to Reviewer 1 and we agree that our findings are consistent with theory. We believe there is value to empirically testing a trend that, as Reviewer #1 notes, has been widely noted anecdotally for particular watersheds or cities. Our results provide empirical evidence to confirm anecdotal and theoretical discussions in the literature. We have also improved the clarity of our discussion to better highlight this interpretation of the results.

1.2: Although the topic is interesting, the contribution of this study to the existing literature is questionable. Only 13 percent of the 416 cities (n=53) included in the analysis actually implement payment schemes. No control is included, for example, for the age of the payment schemes or their institutional-economic design.

This study is one of the first to quantitatively survey cities globally and their use of the IWS strategy. As such, we think it is a noteworthy finding that only 13% of large cities globally have IWS schemes is a result of our study, not a limitation of our sample design. Moreover, we selected a statistical model, random forest, that can be used to model rare occurrences (Boulesteix et al. 2012; Khalilia et al. 2011). We also selected a variable importance measure that is particularly robust against unbalanced datasets (Janitza et al. 2013).

We agree with the reviewer that it would be interesting to describe the age and institutional-economic design of IWS programs, but believe it is beyond the scope and aims of the current paper. Our analysis was designed to investigate those variables assessed by Huber-Stearns et al. (2017) to be enabling conditions for Payments for Ecosystem Services approaches. We would encourage addressing the types of variables discussed by Reviewer #1 in future studies and have added information in Lines 293-294, and 299-311, regarding potential next steps for this research. For this study we attempted to collect data for all cities globally, which is closer to an analysis of an entire population rather than a sample of a population.

Boulesteix, A.L., Janitza, S., Kruppa, J. and König, I.R., 2012. Overview of random forest methodology and practical guidance with emphasis on computational biology and

bioinformatics. Wiley Interdisciplinary Reviews: Data Mining and Knowledge Discovery, 2(6), pp.493-507.

Huber-Stearns, H., Bennett, D., Posner, S., Richards, R., Fair, J., Cousins, S. and Romulo, C., 2017. Social-ecological enabling conditions for payments for ecosystem services. Ecology and Society, 22(1).

Janitza, S., Strobl, C. and Boulesteix, A.L., 2013. An AUC-based permutation variable importance measure for random forests. BMC bioinformatics, 14(1), p.119.

Khalilia, M., Chakraborty, S. and Popescu, M., 2011. Predicting disease risks from highly imbalanced data using random forest. BMC medical informatics and decision making, 11(1), p.51.

1.3: The title of the paper does not reflect its content. Often, payments for watershed services serve multiple goals, not only environmental ones (e.g. Bulte et al., 2008; Brouwer et al., 2011). No control is included for this either.

We considered several different titles and decided on one that we feel accurately and concisely reflects our study, scope, and findings. Our goal was to evaluate the state of IWS programs in a broad sense for large cities globally and analyze which conditions suggest whether IWS could be expanded to additional cities. It's true that many IWS programs serve multiple goals beyond environmental ones (for example, South Africa's Working for Water program, which achieves environmental goals but also serves essentially as a jobs program, among other social functions). An analysis of program goals would be a very interesting next step to this research, but it is not within the scope of this current study, which was focused on evaluating the relative importance of potential enabling conditions synthesized by Huber-Stearns et al. (2017).

Recent literature has increasingly adopted the terminology IWS which is more specific to water quality and quantity related programs, and at the same time for inclusive of the myriad types of investments (not just direct payments) that are occurring in these types of programs. This is in line with some of our data sources (e.g. Bennett & Carroll, 2014; Bennett & Ruef, 2016), and with other recent literature (Huber-Stearns & Cheng 2017, Gartner et al., 2017; any other team members publish using IWS recently?). IWS is also a commonly understood and accepted term within the broader field of PES (which is not water specific).

1.4: It is also unclear how cities were matched, i.e. made comparable, in order to isolate the impact of the 17 control variables included in the statistical analysis.

For this analysis, data were collected for cities that matched a certain criteria (the UNPD definition of "large") and the analysis aggregated these cities as comparable to draw general conclusions. Our analysis here is fundamentally descriptive- it evaluates which of a set of potentially explanatory variables correlates with (or statistically predicts) the presence of IWS schemes.

1.5: Almost a quarter of these 17 control variables furthermore seem to have switching signs (+/-), including the share of protected area in the non-USA cities model (Table 1).

Thank you for your comment and we agree that this type of symbol does not clearly portray the interaction. We have clarified the language for the figure label and changed the signage for the complicated interactions to (~).

1.6: What is more of interest in this field is the evaluation of the incremental impacts of payment schemes for ecosystem services compared to other policy instruments, such as regulation (see the 2015 publication by Naeem et al. in Science about this), or the cost-effectiveness of this instrument compared to traditional engineering solutions to secure safe drinking water supply. This paper does not look at this ("Here we focus on the emergence of IWS programs for cities rather than an evaluation of their impacts"), whereas cost-effectiveness is expected to be one of the key driving forces behind the implementation of this policy instrument. Their emergence as a novel market based instrument has been described in various papers, special issues in journals and textbooks about ecosystem services since 2000. User financed schemes which reduce reliance on limited government budgets for environmental protection and conservation have been propagated, especially in the developing world, as an important impetus behind their emergence (e.g. Wunder, 2015; Engel et al., 2008). Interestingly in this study, however, GDP per capita has a positive effect on their emergence (higher income countries applying payment schemes more likely, in the non-USA model), while conservation spending has a negative effect (higher conservation budgets reduce the need for payment schemes, in both models). No further discussion or explanation is given, nor is there any link provided to the literature in this field. One could argue that the value added of this study is that it tries to quantitatively identify the significance of the role of various driving factors. The question is whether the high level of analysis and the chosen global variables are able to capture the relevant socio-economic, socio-cultural and governance dimensions.

We agree with the reviewer's assessment of the literature. In a previous, related publication on payment for ecosystem programs (PES), all three of the references mentioned were included (Huber-Stearns et al. 2017). In this study, we built from that previous deep review of PES literature and decided not to repeat what others have described. We also took the opportunity to use a quantitative approach to study one particular policy instrument, the IWS program. We agree that it would be quite interesting to do a comparative analysis among different policy approaches, but felt that such an analysis was beyond the scope of our study and the data we had available. Our analysis of non-USA cities identified per capita GDP as positively associated with IWS presence, while conservation spending was negatively associated. To address the reviewers

comments, we expanded discussion on this topic in Lines 277-283 to include the following: This could potentially suggest that a certain level of affluence is needed for IWS and that water providers are not able to spread the cost to users when users are predominately poor. For conservation spending, it may mean that when spending is high, there are co-benefits to water quality coming from other investment actions that make IWS less necessary, similar to our interpretation for the negative relationship with the percentage of protected areas in the watershed.

Where information on the cost effectiveness of various alternative strategies (including IWS) is available, it would enhance understanding of program emergence and sustainability. We include a sentence and citation in Lines 113-115 regarding this potential relationship; Average Annual Growth (the average annual growth rate of national GDP for 1994-2014) also ranked important in both models, possibly because economic growth may increase the resources available for payment for ecosystem services programs and rapid economic growth can increase impacts to water supplies from increased development without infrastructure and institutions in place to address these impacts.²⁴

24. Goldman-Benner, R. L. et al. Water funds and payments for ecosystem services: practice learns from theory and theory can learn from practice. *Oryx* 46, 55–63 (2012).

1.7: I'm not familiar with the applied statistical model, but it would have been good to test the sensitivity of the results to the applied statistical model. As a model robustness check, a simple probit or logit model could have been estimated, for example, regressing the variables representing enabling conditions on the binary variable identifying whether a city has a payment scheme in place.

We agree with the reviewer that a logit or probit model is more commonly used in this field and is a robust analytical model for assessing variables against a binary outcome. Initial attempts at logistic regression resulted in perfect separation and we elected to use a different approach. We were also concerned about the high correlation between some variables and how interactions between these variables would need to be specified. As the first quantitative analysis for these variables in this context, those relationships have yet to be assessed. We believe that our analysis now sets the stage for considering the relationships of the variables most important to predicting the presence of IWS programs. The predictive ability of the model was assessed using AUC (area under the receiver operating curve), which is the standard evaluative procedure for random forest that is robust to unbalanced datasets (Calle et al. 2011). While comparing random forest to probit or logit models would be an interesting exercise, model evaluation was not within the scope of this research. We have included text regarding preliminary logit regression modeling in Lines 66-67, 347-349, and 358-36.

Calle M, Urrea V, Boulesteix AL, Malats N: AUC-RF: A new strategy for genomic profiling with random forest. *Hum Hered.* 2011, 72 (2): 121-132. 10.1159/000330778.

1.8: In conclusion, I don't believe this paper adds enough to the existing literature to warrant publication in Nature. The topic is interesting and the quantitative approach commendable, but the model is too high level and the comparability of the cities and 53 schemes across the implementing and non-implementing cities questionable. More interesting is the question what the impacts are of existing payment schemes on environmental and socio-economic conditions and to what extent these impacts are driven by enabling conditions, including their institutional design, and how cost-effective they are in inducing the necessary behavioral change in the linked urban-agricultural landscape.

This paper aims to provide one of the first global surveys of which large cities are using IWS programs, and describe the statistical predictors of the presence of such program. Reviewer 1 justly describes critical and interesting questions for this field, especially regarding current practitioners or program developers. The questions that Reviewer 1 raises pertain to conservation program design and program impact. However, this is outside the scope of our single study, which is high level because of the global nature of our research questions that explore which enabling conditions could explain the emergence of IWS programs in large cities around the world. Moreover, there simply are not global datasets of some of the variables suggested for study by Reviewer 1, making them difficult to use in a global study such as ours.

Regarding our contribution to the existing literature to warrant publication in Nature Communications, we would like to reference Comments 3.1 by Reviewer 3 and 2.1 by Reviewer 2. . We agree with Reviewers 2 and 3 that this study is worthy of publication in this journal because it is the first global survey of the state of use of urban IWS and provides one of the first quantitative analyses of what factors appear to correlate with or predict the presence of IWS programs. As described by Reviewer 3, this study provides an important significant advancement to IWS program research by evaluating what Reviewer 1 describes in comment 1.1 (repeated here with italics; *These results seem somewhat obvious: the higher the threat (agricultural area), the more likely action will be taken to protect urban water sources, while designated protected areas are simply an alternative policy instrument, reducing the need for additional action such as a payment scheme for watershed services.* Thus far, this relationship has only supported by anecdotal evidence and our analysis provides the first global synthesis of data corroborating that assumption.

Additionally, we address the need for more quantitative evaluation of conditions related to establishment and presence of PES programs. This was articulated in previous work on enabling conditions for PES (Huber-Stearns et al. 2017) and also specifically in a recent Nature paper calling for more documentation and evaluation of external factors that may influence PES programs (Naeem et al. 2015). These types of broad assessments are critical for the relatively new field of ecosystem services to provide more science-based analysis, which our research does. Our hope is that this study sets the stage for further research of the type described by Reviewer 1 to support evidence-based decision-making.

Naeem, S., Ingram, J.C., Varga, A., Agardy, T., Barten, P., Bennett, G., Bloomgarden, E., Bremer, L.L., Burkill, P., Cattau, M. and Ching, C., 2015. Get the science right when paying for nature's services. *Science*, 347(6227), pp.1206-1207.

Reviewer #2 (Remarks to Author)

2.1: The manuscript "Global State and Potential Scope of Investments in Watershed Services for Large Cities" builds an impressive database of cities around the world to better understand what conditions lead to implementation of investment in watershed service (IWS) programs. It uses a random forests classification algorithm to parse out which variables in their dataset are most associated with IWS programs. The manuscript is well structured and is an interesting approach to look for general associations of where IWS programs have sprung up. To my knowledge no other work has compiled such a dataset nor attempted this kind of an analysis.

Thank you for your comments.

2.2: There are areas where the paper could be improved or clarified. One theme throughout my comments is whether the authors can say more about contextual conditions, or how some factors may be important conditional on other factors, which is what we might expect when implementing policy in complex social-ecological systems. The authors present only the globally significant results which, in my mind, mask what could be a much richer exploration of what conditions matter when, and for whom. Without providing some examination of how context matters, the main message falls a bit flat to me.

This comment is very similar to comment 2.3 below. We agree with the reviewer that contextual conditions are critical. We articulate the need for contextual conditions in the discussion and have expanded our thoughts in Lines 303-315 and have provided a more thorough discussion regarding the need for contextual conditions in our response to comment 2.3.

GENERAL COMMENTS

2.3: 1. The overall approach taken by the authors is one that assumes there are generalizable conditions that enable or encourage IWS programs to start in a given location. The authors state that conditions associated with IWS programs "have not been analyzed in aggregate or evaluated for relative importance" (line 53) and that we "need to evaluate the factors associated with IWS programs across many contexts" (line 55). However, why would we expect a consistent set of conditions to be enabling across such a variety of different contexts? Research focusing on program or policy implementation in social-ecological systems shows just how much context and local conditions matter for successful outcomes (e.g., Leslie et al. 2015, Alexander et al 2016, etc.). It would be helpful for this approach to be justified.

We agree with Reviewer #2 that local conditions and context matter, but there still may be general conditions that can make it more likely that a program will emerge. We would expect a

consistent set of conditions to be enabling across a variety of different contexts because others (Waite et al. 2014; McKenzie et al. 2014; Ruckelshaus et al. 2015) have suggested that there are general conditions that can make it more likely that a program will emerge or succeed. We designed this study to test that hypothesis. Additionally, the random forest methodology need not assume a consistent relationship between an explanatory and predictive variable across its whole range, but allows the sign and magnitude of the relationship to vary depending on combinations of other variables. Although the output of the random forest model is a generalization, we explore contexts of individual variables via the partial dependence plots provided as Supplementary Information. These plots describe the relationship of a single variable as the values of that variable change, all other conditions being equal. We have added Box 1 as an example of both how to interpret these plots and also how to implement our findings.

It is useful to understand the general conditions in order to explain (at least partially) the existence of programs and evaluate the potential scope for expanding programs. We articulate the need for contextual conditions in the discussion and have expanded our thoughts in Lines 303-315. Another aspect of our findings relates to the significance of what we didn't find. For instance, some scholars have suggested that strong governance factors need to be in place for IWS to work (e.g., to enforce agreements). However our results show that the governance bar may actually be fairly low, suggesting that IWS could have greater viability in regions where some assume that it wouldn't work. In these situations, one of our findings is that some factors may not be generalizable. We have clarified this aspect of the work in Lines 276-287.

McKenzie, E., Posner, S., Tillmann, P., Bernhardt, J.R., Howard, K. and Rosenthal, A., 2014. Understanding the use of ecosystem service knowledge in decision making: lessons from international experiences of spatial planning. *Environment and Planning C: Government and Policy*, 32(2), pp.320-340.

Ruckelshaus, M., McKenzie, E., Tallis, H., Guerry, A., Daily, G., Kareiva, P., Polasky, S., Ricketts, T., Bhagabati, N., Wood, S.A. and Bernhardt, J., 2015. Notes from the field: lessons learned from using ecosystem service approaches to inform real-world decisions. *Ecological Economics*, 115, pp.11-21.

Waite, R., Kushner, B., Jungwiwattanaporn, M., Gray, E. and Burke, L., 2015. Use of coastal economic valuation in decision making in the Caribbean: Enabling conditions and lessons learned. *Ecosystem services*, 11, pp.45-55.

2.4: 2. Relatedly, it seems there may be more information that the authors could present related to the interactions of contextual conditions. In the methods the authors say they "build a model using interactions between variables" (line 304) but these interactions are not presented or discussed in the paper as far as I can tell. How do interactions matter? What do they reveal about potential factors that matter conditional on other factors?

We have revised the methods description to reflect that in this approach interactions are permitted between parameters, but are considered via the model in their entirety, and not

interpretable on an individual basis. The performance of individual predictor variables within the model context are provided by Supplementary Materials Figure 1 and in the discussion of directionality and behavior (Lines 168-177). We have also added Box 1 to describe implementation and use of these results and SM Figure 1 in particular.

2.5: 3. *The authors use random forests algorithm to analyze the dataset on urban IWS programs. However, little justification is provided for why this method was chosen. What are strengths and weaknesses of this versus other methods like multiple linear regression (e.g., Oliveira et al. 2012, Smith et al. 2013) or maybe even qualitative comparative analysis (QCA) (Ragin 2007)?*

This comment is similar to comment 1.7 regarding the use of regression models. We also partially address this comment in our General Changes. We have edited the manuscript to include more information about the modeling algorithm and justification for the model in Lines 66-67, 347-349, and 358-36.

2.6: 4. *Admittedly, I have only casual knowledge of random forest algorithms, so my perspective on methods is one of a slightly informed but somewhat lay reader. In this regard, I am used to seeing decision trees as an outcome of classification methods like random forests, which often at least helps show how splits in the data reveal different conditional patterns and processes, which may relate to different contexts and provide some evidence that the authors call for in the discussion (lines 241-252). However, in this manuscript (throughout the main text and the supplementary materials) the authors rely only on the variable importance diagram (Figure 3) as the main outcome from which they rest all their results and interpretation. Why have the authors chosen to rely on such a limited set of results?*

This comment is similar to comment 1.7 regarding the use of regression models. We also partially address this comment in our General Changes. We have edited the manuscript to include more information about the modeling algorithm and justification for the model in Lines 66-67, 347-349, and 358-36. Here we add to our previous responses to address specifically the use of a single classification tree. Random forest was used instead of a single classification tree because we are interested in the typical outcome of many trees, not the probabilities that are related to each node and branch. The amount of highly correlated predictor variables can be handled by Random Forest, but would not allow for a robust analysis using a single classification tree or regression model. Initial attempts at logistic regression resulted in perfect separation and we elected to use a different approach.

2.7: 5. *The authors rightly note that their analysis would need to be supplemented with other information to inform actual decision-making. But the limitations to this "technocratic" approach could be better highlighted. Namely, that it does not take into account local economic (what are the costs and benefits of a program in a given place) (Zheng et al. 2013), political, or social conditions that may affect the appropriateness of a program.*

In response to this comment, we have elaborated on the need further analyses regarding other

local considerations that examine the cost effectiveness of the program as well as cultural appropriateness. We also have included a box that provides explanation of how to apply our model to management (Lines 303-315).

EDITORIAL COMMENTS

line 66

2.8: The title of this section is "Global and Regional Distribution of IWS" which implies the sample is representative and that this is, indeed, indicative of the distribution of IWS. However, no information is given as to whether the "stratified sample" (line 68) developed by McDonald et al was random or representative, nor the sampling procedure for the other 59 cities the authors "identified" (line 69). Were these randomly sampled? Does the reader have any assurance these are indeed representative of global distributions? If not, its hard to reliably interpret any of the regional breakdowns that are presented.

This is a good point regarding information flow for readers. The reviewer comment is in the Results section and the pertinent information regarding representation can be found in the Supplemental Methods. We've added clarification to the Methods section of the main manuscript body and also added some information about the methodology to the Introduction.

2.9: lines 84, 90

the use of the word "dataset" is confusing here, as it seems in the text the authors are referring to just one variable from a single dataset.

Similar to other comments, we have clarified that individual predictor variables (data) were extracted from existing datasets and databases. Thank you.

2.10: line 102

suggest replacing "not important" with "not statistically significant at a 0.05 level of confidence." This is not the same as importance.

We agree that "not statistically significant at a 0.05 level of confidence" is not the same as importance. The modeling method used in this analysis is a machine learning algorithm that does not incorporate frequentist statistics and presents confidence levels and significance differently. This model is a type of selection process that compares variables. The appropriate terminology for Random Forest is "importance" or "variable importance" per Breiman 2001. We selected this algorithm specifically for robustness in dealing with the type of data we have and allows us to rank variables in terms of importance in predicting the presence of absence of IWS.

Breiman, L., 2001. Random forests. Machine learning, 45(1), pp.5-32.

2.11: line 121

how did breaking the data into two models confirm the authors' "understanding of how the random forest algorithms identify important predictors"? What confusion did the author's have

without this confirmation?

Anticipation of this relationship partially motivated our choice to build the Non-USA Cities model, and observing the differences in enabling conditions between models confirmed our understanding of how random forest algorithms identify important predictors. We have clarified this within the text in Lines 127-131.

2.12: line 137

To the reader its unclear how the partial dependence plots show how the variable relates to whether there is or is not an IWS program. This is a binary condition, but the plots are continuous. I think the authors mean something like: These plots indicate the probability of that variable being present in cases where an IWS program is present relative to cases without an IWS program.

Thank you for your comment, we agree that the use of partial dependence plots needed clarification for the reader. The plots reveal how well different values of a particular variable predicts the outcome when all other variables are held constant. We believe this information to be important because in many cases, marginal shifts at low values result in large changes in predictability. This could be operational in the case of comparing two cities that had similar values for one variable, but different values for another variable. How the differing variable performed in the partial dependence plots provides information about whether an IWS program is present and potentially whether an IWS program could be instated successfully. We have clarified this information in Lines 169-178.

2.13: line 139

Does “the outcome” refer to the presence or absense of an IWS program?

Yes. We have added "presence or absence of an IWS program" as parenthetical clarification in Line 172.

2.14: line 140-141

how much confidence can we have in such statements? Can the authors include confidence intervals around these probabilities in the partial dependence plots?

The partial dependence plots depict how much influence a particular variable has in predicting the outcome within the context of our model. We have included clarification about this relationship in Lines 169-178.

2.15: line 319

Supplementary Table 3 seems to be missing

Supplementary Table 3 will be included with the revised manuscript.

2.16: Table 1

some of the variables (e.g., “IUCN Organizations Per Million People” and “Registering

property”) are repeated across multiple conditions. Does this mean the variable is repeatedly included or just that this single variable represents several different governing conditions? This is potentially confusing to the reader.

We appreciate this comment and have revised Table 1 and the caption to make it easier for the reader to understand these results. The repetition of multiple variables across conditions reflects the potential for the variable data to represent different conditions simultaneously. This is addressed in the revised Table 1 by organized rows by predictor variable. In this way, each predictor variable appears once, but may be associated with multiple enabling conditions in the 5th column. A revised version of the original Table 1 is provided in the supplemental information to provide information regarding which enabling conditions were tested and what data were considered representative of each enabling condition.

2.17: Figure 3

Indicate the meaning of the red dotted line in the table notes or on the figure directly.

We have added a sentence explaining the red dotted line.

2.18: Supplementary Figure 1

** Label x axis in these plots. Is this the range of the data for a given variable? The y axis is probability of what?*

** The Figure caption refers to Figure 3. There are no lines in figure 3 of the main text? Or do the authors mean this figure (Supp Fig 1)?*

** The Figure caption says "Line shapes in Figure 3 explain directionality of prediction ability and thresholds, but scale and numbers do not represent accuracy or prediction". It is not clear to me whether this is supposed to refer to Figure 3 or Supplementary Figure 1, nor is it clear to me what this phrase means.*

** The Figure caption notes the variables are displayed in rank order. Indicate whether this is left to right or top to bottom.*

Thank you for your careful consideration of Supplementary Figure 1. We have made changes to per the Reviewer's recommendations and to clarify how to interpret this figure.

2.19: Supplementary Table 2

Formatting — the table is split into two pages, I assume the authors intended this to be in landscape format.

Thank you for the comment. Reviewer #2 is correct in that the table is intended to be presented in landscape format.

2.20: Supplementary Methods (SM)

Sometimes the list of figures and tables here relative to the main text is confusing. I recommend

that all figures and tables in the SM document be listed as SM Table 1, SM Table 2, etc. to avoid confusion.

This is a good idea. We have revised the list, thank you.

Reviewer #3 (Remarks to Author)

3.1: The paper offers an interesting quantitative analysis of enabling conditions for watershed conservation schemes for large cities. It is of critical importance to gain a better understanding of the factors important for emergence of water services protection schemes on the ground and the research field has gained momentum in recent years. This is evident from the paper and the authors demonstrate a good overview of the current state of knowledge. The paper is a global analysis, which is the first of its kind according to the authors. Other studies in the literature have tended to focus on individual case studies or a few cases. I agree with the authors that the global scale analysis of payment schemes for ecosystem services is rare in the literature and the paper offers a novel contribution. The paper uses a decision tree approach to the analysis of enabling factors. The method choice appear suitable and is also [sic - reviewer sentence unfinished]

3.2: The paper is well written and clearly organized and appears to follow the journal style and format requirements. I have however some suggestions for improvement of the paper both in terms of clarity and in terms of justification of the approach taken.

Thank you to Reviewer #3 for the compliments regarding organization and writing.

Major comments:

3.3: 1) The caption for Figure 1 states the figure is modified from Huber-Stearns et al. It looks like a copy – at least I can't tell what the modifications are. From the methods section it is clear that there is some overlap between the authors of this paper and the current paper, so that the authors are only copying from themselves. Even so, it is not appropriate to claim that the current paper offers more novel material than it actually does.

Modifications were very minor (color and footnotes), but the reviewer is correct that the figure is essentially the same. We have revised the caption for Figure 1 to indicate that it is essentially unmodified.

3.4: 2) The paper refers to IWS as the core concept. It is not clear to me how this concept differs from PES and why you choose to change the terms used in the literature.

The nomenclature surrounding both these types of programs and the enabling conditions are discussed in Huber-Stearns et al. (2017). We considered several possible terms to describe the programs we studied: payment for ecosystem services, payment for watershed services, water funds, source water conservation, etc. We decided to use "investment in watershed services" in

order to align ourselves with the more recent Forest Trends reports that we used. We also feel that the word "investment" reflects more nuance about the financial transaction than "payment."

Huber-Stearns, H., Bennett, D., Posner, S., Richards, R., Fair, J., Cousins, S. and Romulo, C., 2017. Social-ecological enabling conditions for payments for ecosystem services. *Ecology and Society*, 22(1).

3.5: 3) Table 1: IUCN organisations per million people appears as indicator of three different enablers. While I appreciate that it is difficult to identify very accurate indicators of all the variables indicated in Figure 1 it is misleading that include all the variables in Table 1 when three of them are the same variable. I think it would be better to reduce the number of variables in the table and explain in the text that some of the variables do not clearly represent only one of the types of enablers. This is also the approach the authors take in Figure 3.

This is a similar comment to 2.16 and we have revised Table 1 to provide one row per predictor variable while still providing enabling condition information.

3.6: 4) Methods: The authors say that models are build based on 17 datasets – I interpret this as the different enablers. Is his correct?

Clarified in the text (Lines 59-61) that, using a random forest machine learning algorithm, we tested 17 datasets representing 15 of the 24 identified enabling conditions with some datasets representing a set of enabling condition concepts (Figure 1).

3.7: I get the impression that the difference in the models developed is due to the high correlation between variables. However, I am not entirely sure that this is what happening. What do you use the correlation analysis for? Could you clarify?

Initially we used the correlation analysis to select variables to model from the 30 candidate variables identified that potentially represent enabling conditions. This process is described in the Supplemental Information file under the header "Variable Selection" (starting at Line 139). We believe that the difference in the models is largely driven by the low variation in certain country-level data when USA cities are included. Less variation in a variable would result in a lower importance ranking because it would be harder for the algorithm to differentiate outcomes when many cities have the same variable value. We have changed some text in the Results (Lines 127-131) to describe this situation.

3.8: It is also not clear how you choose the preferred model. Is this based on the AUC ? Is this part of the Random Forest approach or do you select the preferred model?

Preferred models were selected using AUC, though the initial manuscript only had a simple line to describe the process. This model selection method is commonly used for random forest and proven robust to unbalanced datasets (Calle et al. 2011). We have added text in Lines 376-384 to add clarity to our model selection and full model parameters and assessment are described in

the Supplemental Information file (Lines 202-218). We have also included our code as a supplemental html file.

Calle M, Urrea V, Boulesteix AL, Malats N: AUC-RF: A new strategy for genomic profiling with random forest. *Hum Hered.* 2011, 72 (2): 121-132. 10.1159/000330778.

3.9: Furthermore, while the random forest approach appears interesting it is not clear what is gained relative to more simple regression type models predicting the presence of IWS. The authors argue that the random forest approach allows the researcher to capture complex relationships between enabling factors. However, it is not really clear how this strengthen the analysis and findings in the study presented. The supplementary material indicates that the classification is a simple tree with threshold cut-off points to determine probability of a city having a IWS. I have probably misunderstood this, but I think the paper would benefit from a clearer description of how the more complex methods enrich the analysis.

This comment is similar to comment 1.7 regarding the use and justification of the model, and is also similar to comment 2.6 regarding other regression models. We also partially address this comment in our General Changes. We have edited the manuscript to include more information about the modeling algorithm and justification for the model in Lines 60-69 and 362-365. We have also included the raw data as a supplemental table and our statistical code as a supplemental html file. Initial attempts at logistic regression resulted in perfect separation and we elected to use a different approach. We have included text regarding preliminary logit regression modeling in Lines 66-67, 347-349, and 358-36.

3.10: 5) I like Figure 3 but it is not clear to me how to interpret the values on the x-axis. Could you explain?

The Reviewer rightly points out that the interpretation of these figures is not intuitive and we have added clarifying text in the Figure label.

3.11: 6) It is not clear to me what you use the partial plots for. The authors refer to a Figure 3 in the supplementary material that would probably help but the Figure does not appear to be submitted?

"Figure 3" label removed; language applies to Supplementary Figure 1. We have also added a Box to the main manuscript that explains implementation of results, using information from this figure.

3.12: Supplementary material:

The supplementary Table 1 gives the description of how the enabling factors have been operationalized in the quantitative analysis by outlining the variables chosen and the data sources. This is very helpful and gives a clear overview of how the authors have ended up with due to data shortage at the global scale.

We are glad this was useful.

Supplementary material table 2:

3.13: Format of the supplementary material – Table 2 – needs reformatting, as it is not possible to see which cities link to some of the enablers.

Applied formatting such that odd numbered rows have grayed text.

3.14: The caption in table 2 refers to partial plots in supplementary material Figure 1. This Figure does not appear to be in material ?

The Supplementary Figure 1 is found on the first page of the Supplementary Information file.

Minor comments:

3.15: Line 27: Not clear what you mean by water risk.

Clarified sentence: Management innovations have begun offering solutions to problems pertaining to water scarcity, quality, and risk threats to availability.

3.16: Line 159: A double full stop.

Fixed, thank you.

3.17: Line 193: A double full stop.

Fixed, thank you.

Reviewer #2 (Remarks to the Author):

The revision of the manuscript "Global State and Potential Scope of Investments in Watershed Services for Large Cities" addresses many of the original comments. The discussion around local/contextual conditions is improved, and the addition of Box 1 is nice.

It is also good to see that the authors include some reference to logistic regression approaches, in addition to the authors' preferred Random Forests (RF) method. However, I have two issues with their justification in the manuscript.

First, the authors justify use of RF by saying "random forest also reduces issues of bias toward the majority class that occurs with unbalanced datasets in logistic regression". However, for logistic regression imbalance is not generally an issue — there is not "bias" toward the majority class per se. From my understanding in the literature, the main issue with using logistic regression for unbalanced data has more to do with whether there is enough variation in the minority class to enable accurate predictions. Its not clear to me that RF handles this better (though I am admittedly not an expert on RF)? The single source the authors use to justify RF over logistic regression makes convincing arguments that RF is better (a) when the imbalance is on the order of 100:1 [in the current manuscript the ratio is about 7:1 (14%)] and (b) for prediction of the outcome of interest [from what I understand the authors are mainly concerned with identifying factors that are important, not with the model's predictive ability per se].

Second, the authors solely justify the use of RF based on the idea that their dataset is unbalanced. Yet in the comments to reviewers they note a number of other reasons (higher-order interactions, nonlinear relationships among predictors, etc.) but these are not flushed out or noted in the main manuscript. As I note above, I do not find the justification on imbalance (alone) very compelling.

Model choice and justification is important to ensure the robustness of the results, as the two other reviewers also note. It seems to me that in this data set robust relationships should surface in both approaches. I was shocked that the author' suggest in their response to reviewer point 1.7, that "...model evaluation was not within the scope of this research". This is simply not an excuse.

If perfect separation is a problem, then at least devote some supplementary material to this: to which variables does the model perfect separate (I assume/hope the same ones that are "important" RF outcomes?) If that variable is removed, what else appears important? If multicollinearity is a problem, what factors are important with a more parsimonious use of the data?

RESPONSE TO REVIEWERS

Re: manuscript titled *Global State and Potential Scope of Investments in Watershed Services for Large Cities*

We thank Reviewer 2 for their careful consideration and second round of review. This second round of revision has helped strengthen our manuscript further and provide a higher quality presentation of the work. In this letter we detail how we have addressed all comments. Reviewer comments are provided in *italicized text*. Throughout, line numbers in referee comments refer to the original manuscript and line numbers in responses refer to the revised manuscript. We have numbered comments to indicate both the reviewer and comment number (2.X indicating all comments from Reviewer #2). Changes from the previous revision were accepted and all new changes to text in the manuscript file are presented with track changes. New citations have been highlighted yellow in the revised manuscript.

Reviewer #2 (Remarks to Author)

2.1 The revision of the manuscript "Global State and Potential Scope of Investments in Watershed Services for Large Cities" addresses many of the original comments. The discussion around local/contextual conditions is improved, and the addition of Box 1 is nice.

Thank you for your comment.

2.2 It is also good to see that the authors include some reference to logistic regression approaches, in addition to the authors' preferred Random Forests (RF) method. However, I have two issues with their justification in the manuscript.

First, the authors justify use of RF by saying "random forest also reduces issues of bias toward the majority class that occurs with unbalanced datasets in logistic regression". However, for logistic regression imbalance is not generally an issue — there is not "bias" toward the majority class per se. From my understanding in the literature, the main issue with using logistic regression for unbalanced data has more to do with whether there is enough variation in the minority class to enable accurate predictions. Its not clear to me that RF handles this better (though I am admittedly not an expert on RF)? The single source the authors use to justify RF over logistic regression makes convincing arguments that RF is better (a) when the imbalance is on the order of 100:1 [in the current manuscript the ratio is about 7:1 (14%)] and (b) for prediction of the outcome of interest [from what I understand the authors are mainly concerned with identifying factors that are important, not with the model's predictive ability per se].

The reviewer brings up a very good point regarding how to address data imbalance. We agree that the main issue specific to data imbalance is whether sufficient variation exists to enable prediction, and that researchers should be careful in situations with large imbalance because outcomes can be predicted by randomly selecting the majority class, rather than relying on all

available data. This is a concern we take very seriously and also the reason why we present two separate models of the research - one inclusive of all cities, and the other subsetting non-USA cities only. Some of our datasets only provide information at the scale of a country. With so many cities in the USA, these cities all receive the same value for country-level data, which would decrease the ability of these variables to explain variation in the data and result in artificially lower importance rankings. Random forest methods also reduce issues of bias toward the majority class that can occur with unbalanced datasets in logistic regression (Muchlinski et al. 2016), important because in this dataset cities with IWS programs represent the minority class. There were not enough large U.S. cities with IWS programs in our dataset to justify a random forest model of only cities in the United States. This explanation of the two models is provided in lines 120-127, and also described in the Supplementary Information.

In this new revision, supplemental material is provided to the reviewers as a shiny app site and will also be made available to readers for viewing data and descriptive statistics. This link has had author identifying information removed, but that information will be replaced in the final manuscript if accepted for publication.

Link: <https://iwsprograms.shinyapps.io/InvestmentInWatershedServices/>

As can be viewed by the plots and histograms in the supplemental tables & figures, many of our predictor variables have overlapping ranges of values for the two dependent variables (cities with IWS programs and cities without IWS programs). Many predictors also present with non-normal distributions such as Poisson or Bernoulli. Some also consist of categorical data or nominal data in addition to continuous data. The random forest model allows us to consider all variables together and determine which are important to determine outcomes.

We have added clarification where we introduce RF as our methodology in the Introduction section (changes in Lines 52-85), including an expanded description of our research objective in Lines 58-64 to better clarify our analytical needs. We have added explanation of data characteristics in Lines 78-85 and 367-368 and also describe how RF is well-suited to handle these data situations (especially in Lines 78-85).

Muchlinski, D., Siroky, D., He, J. & Kocher, M. Comparing Random Forest with Logistic Regression for Predicting Class-Imbalanced Civil War Onset Data. *Polit. Anal.* 24, 87–103 (2016).

2.3 Second, the authors solely justify the use of RF based on the idea that their dataset is unbalanced. Yet in the comments to reviewers they note a number of other reasons (higher-order interactions, nonlinear relationships among predictors, etc.) but these are not flushed out or noted in the main manuscript. As I note above, I do not find the justification on imbalance (alone) very compelling.

We have made it more clear that we're justifying the use of RF based on more than solely an imbalanced dataset -- as the reviewer notes, we use RF based on "higher-order interactions, nonlinear relationships among predictors, etc". We've added more text in the main manuscript in Lines 78-85 to provide this justification to readers, including additional citations to other notable research that use this modeling system to answer similarly structured research questions.

2.4 Model choice and justification is important to ensure the robustness of the results, as the two other reviewers also note. It seems to me that in this data set robust relationships should surface in both approaches. I was shocked that the author' suggest in their response to reviewer point 1.7, that "...model evaluation was not within the scope of this research". This is simply not an excuse.

We agree with the reviewer that model choice and justification is critical and we appreciate the opportunity to provide more robust explanation of our selection of a random forest approach. To clarify our previous point regarding model evaluation, our choice was based on fitting both our research objectives and characteristics of the data. We have added clarifying text in lines 76-86 regarding the specific justification for this approach. Evaluation is absolutely a critical and important step for demonstrating robustness; we rely on established methods suited to justification of our selected model rather than a comparison of various modeling methods.

Random forest model predictive validation is assessed via AUC, which is a standard metric used for these type of machine learning ensemble methods^{1,2}. A good example of applying a similar method (boosted regression trees) to global prediction with model assessment using AUC was recently published in Nature Communications³. We provide explanation of this assessment in Lines 375-379 in the main manuscript and also Lines 210-218 of the supplemental information.

1. Hosmer, D.W., & Lemeshow, S. *Applied Logistic Regression* (2nd Edition). New York, NY: John Wiley & Sons (2000).
2. Strobl, C., Boulesteix, A.-L., Zeileis, A. & Hothorn, T. Bias in random forest variable importance measures: Illustrations, sources and a solution. *BMC Bioinformatics* 8, 25 (2007).
3. Allen, T. et al. Global hotspots and correlates of emerging zoonotic diseases. *Nat. Commun.* 8, 1124 (2017).

2.5 If perfect separation is a problem, then at least devote some supplementary material to this: to which variables does the model perfect separate (I assume/hope the same ones that are "important" RF outcomes?) If that variable is removed, what else appears important? If

multicollinearity is a problem, what factors are important with a more parsimonious use of the data?

We agree that it's important to examine the data with different variables removed to understand how more parsimonious use of the data might change our results. We include the Shiny App link above to allow exploration of descriptive statistics for each variable in our dataset. However, we emphasize the value of our focus on the **relative** importance of the many variables when analyzed together. In this way, our results improve our understanding of this topic by showing "Situations where certain variables are more important than others" rather than "Variable A is important because it's related to IWS on its own" or "Variables A and B, but not C, are important when all other variables are removed." Our use of Random Forest to examine this unique dataset provides novel insights into the relative importance of these variables. The exploration in relative importance reflects an industry need from our collaborators at The Nature Conservancy, who are quite familiar with those variables which are distinctly important or related to IWS presence, but struggle with new site selection given the amount of potentially important variables that could be considered. Evaluating relative importance allows reflection of not just what is important individually, but which important variables deserve attention, as we outline in Box 1.